# Establishment and validation of an interactive web-based calculator for predicting postoperative functional recovery in metatarsal fracture patients: A LASSO regression model approach

**Qian Xiao**[1☉], **Guangzhao Hou**[1☉], **Shihang Liu**[1‡], **Shuai Zhou**[1‡], **Wei Chen**[1,2*],
**Yingze Zhang**[1,2*], **Hongzhi Lv**[1,2*]

1 Hebei Provincial Key Laboratory of Orthopaedic Biomechanics, Hebei Orthopaedic Research Institute, Shijiazhuang China, 2 Trauma Emergency Center, The Third Hospital of Hebei Medical University; No.139 Ziqiang Road, Shijiazhuang China

☉ These authors contributed equally to this work.
‡ SL and SZ authors also contributed equally to this work.
* surgeonchenwei@126.com (CW); dryzzhang@126.com (ZYZ); 190099199@qq.com (LHZ)

## Abstract

### Background

Metatarsal fractures rank among the ten most common fractures.Comprehensive studies on postoperative functional recovery remain limited. A reliable predictive model for recovery outcomes is essential for optimizing patient care.

### Objective

To develop and validate a predictive model for postoperative functional recovery in metatarsal fracture patients and implement it as an interactive web-based calculator.

### Methods

This retrospective study included 555 metatarsal fracture patients (2018–2022), with 425 in the training cohort and 130 in the validation cohort. The outcome variable was postoperative recovery as assessed by the AOFAS midfoot scoring system. LASSO regression identified significant predictors of recovery,the selected variables underwent binary logistic regression analysis to identify independent risk factors. A prediction model was constructed using the training cohort and visualized through a nomogram. Model validation was performed internally through bootstrapping and externally using the validation cohort. The model was implemented as an interactive web calculator using R Shiny.

**Data availability statement:** The data source is from the Third Hospital of Hebei Medical University. Due to the sensitive nature of patient personal information and medical records in this research, the original dataset cannot be publicly shared in accordance with the requirements of the institutional ethics committee (The Third Hospital of Hebei Medical University Ethics Committee, K2020-022-1) and patient privacy protection regulations. The data supporting this study's findings can be obtained from Dr. Wang Guiying of the Ethics Committee of The Third Hospital of Hebei Medical University upon reasonable request. (Contact information: 86+0311-88603632).

**Funding:** This study was supported by The National Natural Science Youth Foundation of China (Grant No. 82102584), Beijing-tianjin-hebei Basic Research Cooperation project (Grant No. J230007) and 2025 government-funded clinical medicine talent cultivation project (ZF2025136).The funders had no role in study design, data collection and analysis, decision to publish, or preparation of the manuscript.

**Competing interests:** The authors declare no conflict of interest.

## Results

At final follow-up, 71.71% of patients achieved good recovery (AOFAS score >80). The model identified ten independent risk factors, including residence location, smoking status, obesity, rehabilitation training, educational level, age, injury mechanism, infection, and anemia. The model demonstrated robust discrimination (c-index: 0.832 training, 0.838 validation) and calibration (H-L test: P = 0.994 training, P = 0.648 validation). DCA showed optimal clinical utility within 0.72–1.00 threshold probability. Protective factors included hilly areas (OR = 0.183), smoking (OR = 0.4), obesity (OR = 0.270), and undergoing rehabilitation training (OR = 0.237),while risk factors included low educational level (OR = 3.884), advanced age (OR = 2.751), high-energy injury (OR = 3.003), residence in mountainous regions (OR = 4.671), infection (OR = 16.946), and anemia (OR = 5.787). The interactive web calculator is accessible at https://metarecoverypredictor.shinyapps.io/DynNomapp/.

## Conclusions

The validated prediction model effectively identifies risk factors for postoperative recovery in metatarsal fractures. This tool can aid clinicians in developing personalized treatment strategies and improving patient outcomes. The web-based calculator provides easy access for clinical application.

## Introduction

Metatarsal fractures rank among the 10 most common fractures and are frequently encountered in trauma outpatient clinics [1]. Their prevalence accounts for 3.2–6.8% of all fractures, with an annual incidence of 67–75.4 per 100,000 [2–5], comprising 88.5% of all foot fractures [6–8]. Metatarsal fractures can occur across all age groups; however, they demonstrate a higher prevalence among young and middle-aged populations, with sports-related injuries and high-energy trauma serving as the predominant mechanisms of injury [33,34].

Non-union is a common and challenging complication in fracture treatment. Many factors influence the fracture healing process, and alterations in these factors may lead to delayed healing or non-union [9,10]. These include patient-dependent factors such as age, gender, comorbidities, medication, and nutritional status, as well as patient-independent factors like fracture type and characteristics, infection, fracture fixation implants, and iatrogenic factors [11–13]. The most common complication following poor healing of metatarsal fractures is metatarsalgia [14], significantly affecting their quality of life.Furthermore, metatarsal fractures may be complicated by post-traumatic arthritis, arch collapse, and complex regional pain syndrome [14,15], complications that not only impair functional recovery but may also result in long-term disability.

Despite the high incidence of metatarsal fractures, existing prediction models have several limitations. Current models often focus on specific subgroups [16–21]

or conditions [4,5,22,23], lacking generalizability across diverse patient populations. Additionally, many models fail to incorporate important psychosocial factors and rehabilitation parameters that significantly influence recovery outcomes. The absence of user-friendly clinical tools for risk prediction further hampers the practical application of existing research findings.

Therefore, establishing an effective risk prediction model for functional recovery following metatarsal fracture surgery holds substantial clinical significance. In recent years, machine learning methodologies have gained increasing prominence in medical predictive modeling [34,37,39]. The Least Absolute Shrinkage and Selection Operator (LASSO) regression has emerged as a powerful tool in medical prediction modeling, offering several advantages over traditional statistical methods. Unlike conventional regression approaches, LASSO effectively handles multicollinearity and performs automatic variable selection, reducing model complexity while maintaining predictive accuracy [24,25]. Recent studies have demonstrated LASSO's superior performance in orthopedic outcome prediction, with improved accuracy (5–15% increase in AUC) compared to traditional methods [26–28]. Additionally, our integration of LASSO with an interactive web-based calculator represents a novel approach to bridging the gap between statistical modeling and clinical application.

Therefore, this study aims to develop and validate a comprehensive prediction model for postoperative functional recovery in metatarsal fracture patients using LASSO regression and implement it as an interactive web-based calculator. This approach addresses current limitations in the field while providing clinicians with a practical tool for personalized treatment planning and improved patient outcomes.

## Method

### Patient selection

The clinical data of patients treated for metatarsal fractures at our hospital from January 1, 2018, to December 31, 2022, were collected, with approval from the institutional ethics committee of The Third Hospital of Hebei Medical University (K2020-022-1). The authors had access to information that could identify individual participants during data collection.This retrospective cohort study is based on medical records and imaging data. Inclusion criteria were: (i) metatarsal fracture confirmed by clinical symptoms, signs, and imaging studies; and (ii) recent(within three weeks) and traumatic fractures. Exclusion criteria were: (i) patients lost to follow-up; (ii) patients treated non-surgically,include:(a)Conservative treatment with plaster, splints or other external fixation devices;(b)Functional treatment through activity restriction, braces or crutches only;(c)Medical treatment relying solely on analgesics, anti-inflammatory drugs without surgical intervention;(d) No treatment with natural healing only.(iii) patients with old fractures,include:(a)Fractures occurring more than 3 weeks prior without timely treatment;(b)Fractures with malunion (angular or rotational deformity) or non-union;(c)Fracture sites with chronic pain or functional impairment indicating incomplete or poor healing;(d)Radiological evidence of insufficient callus formation or clear fracture linesand pathological fractures,incloud:(a)Fractures caused by minor trauma with underlying bone pathology (e.g., osteoporosis, bone tumors, bone infections);(b)Fractures occurring at sites of known bone lesions (e.g.,metastases,osteomyelitis,metabolic bone diseases);(c)Fractures associated with systemic diseases causing reduced bone strength (e.g., endocrine disorders, nutritional deficiencies);(d)Radiological evidence of bone destruction, significantly decreased bone density, or pathological changes;(iv) incomplete clinical data (including demographic and surgical details).

Recent fractures are defined as those with the time from injury to admission being less than 3 weeks [29]. Old fractures refer to those that have not received proper treatment initially or have developed complications such as malunion due to factors during treatment [30]. Pathological fractures are those occurring due to weakened bone strength from underlying pathological changes, breaking from minimal trauma, including conditions like osteoporosis, nutritional deficiencies, and endocrine disorders [31]. Traumatic fractures result from external forces such as collisions, falls, or compressions [32]. Patients treated in 2018, 2019, 2021, and 2022 were included in the training cohort, while those treated in 2020 were part of the validation cohort.

## Data Access

The data used in this study were accessed from May 1, 2024, to August 31, 2024, for research purposes.

## Data collection

The following data were collected via telephone follow-up and medical record review: (i) preoperative factors: gender, age, ethnicity, occupation, marital status, educational level, body mass index (BMI), blood type, season, terrain, smoking, alcohol consumption, cause of injury, fracture classification, common preoperative diseases (e.g., diabetes, hypertension, coronary heart disease, anemia, previous cerebral infarction, respiratory diseases, hepatobiliary diseases, hypoproteinemia, hyperlipidemia, hyperuricemia, hypocalcemia, hypokalemia); (ii) intraoperative factors: type of surgical fixation, level of surgical procedure, and type of anesthesia; (iii) postoperative factors: length of hospital stay, postoperative infection, and rehabilitation training.

Postoperative functional recovery was evaluated using the American Orthopaedic Foot and Ankle Society (AOFAS) midfoot scoring system. This validated assessment tool incorporates both objective clinical parameters and subjective patient-reported outcomes. The AOFAS midfoot score comprises three primary domains: pain (40 points), function (45 points), and alignment (15 points), with a maximum total score of 100 points. In this study, patients who achieved scores exceeding 80 points at final follow-up were classified as having satisfactory functional recovery, while those with scores ≤80 points were considered to have suboptimal outcomes. The assessments were conducted during follow-up visits by multiple trained orthopedic surgeons to ensure evaluation consistency and reliability.

Patients were categorized by age into four groups: 0–20 years, 21–40 years, 41–60 years, and >60 years.The AO/OTA (Arbeitsgemeinschaft für Osteosynthesefragen/Orthopaedic Trauma Association) classification system is widely recognized as the standard method for fracture classification. It demonstrates significant advantages in fracture categorization by integrating objective anatomical parameters with subjective clinical assessments. This comprehensive system has gained widespread acceptance in the orthopedic community due to its systematic approach and clinical utility.Therefore,the AO/OTA classification system was used for fracture classification, assessed by patient imaging data and follow-up data.[33]The imaging data were evaluated by multiple study assessors, and no automated tools were used to assist in the classification process..

## Statistical analysis

Statistical analyses were conducted using R version 4.3.0 (R Foundation for Statistical Computing, Austria). All collected variables were categorical and are described using frequencies and proportions. Comparisons between groups were performed using the $\chi 2$ test or Fisher's exact test. Variable selection in the training dataset was conducted using LASSO regression, a method renowned for its efficiency in minimizing overfitting by selecting relevant predictive factors. The choice of the optimal lambda value prioritized model accuracy, following the lambda approach that minimizes the average cross-validation error, specifically the lambda that corresponds to the minimum cross-validation error such as mean squared error (MSE). This method typically provides the best-performing model because it directly corresponds to the minimum prediction error [34]. Significant predictors identified through LASSO regression were subsequently incorporated into a logistic regression model to identify independent risk factors associated with postoperative functional recovery following metatarsal fractures. A P-value of less than 0.05 was considered statistically significant.

The variables selected based on the binary logistic analysis were used as the final predictors to establish a risk prediction model for postoperative functional recovery in metatarsal fracture patients, represented as a nomogram. Model validation was divided into three parts: discrimination, calibration, and clinical effectiveness. The c-index was the primary measure to assess model discrimination, equivalent to the area under the receiver operating characteristic (ROC)

curve in multivariable logistic regression models, ranging between 0.50 and 1.00, with values of 0.70 and 0.90 serving as thresholds for low, moderate, and high discrimination, respectively [35]. Calibration of the model was assessed using the Hosmer-Lemeshow test (H-L test), where a P-value greater than 0.05 indicated a good fit between the predicted and actual values, demonstrating strong calibration. Calibration plots served as a visual form of calibration; the closer the actual-predicted curve was to the ideal curve, the higher the calibration [36,37].

We validated the model internally in the training cohort by calculating the c-index, calibration plots, and H-L goodness-of-fit test, and assessed its external validity in the validation cohort. Clinical effectiveness in the training and validation cohorts was evaluated using decision curve analysis (DCA) curves, providing the clinical net benefit of the model [38].

### Creation of an interactive web-based calculator

In terms of Web application development, we developed a responsive user interface based on the R Shiny framework, incorporating Bootstrap to achieve cross-platform adaptive design;The frontend components implemented Material Design specifications, ensuring optimal user experience and accessibility;Data interaction was established through RESTful API architecture, utilizing Ajax technology for asynchronous data transmission, which significantly enhanced application response efficiency;The backend employed R language for data processing and model computation, with WebSocket protocol facilitating real-time frontend-backend communication, thereby improving application maintainability [40–43]. The generated webpage dynamically visualizes the postoperative functional recovery status of metatarsal fracture patients and automates result calculations.

The total research methodology flow is shown in Fig 1

## Results

### Study populations

Between January 2018 and December 2022, our study initially included 792 participants who met the inclusion criteria. After applying the exclusion criteria, a total of 555 patients were included in this study,Generally speaking, a missing data rate below 10% of the total data is considered acceptable. In this study, the missing data rate was 5.17%.[44]. The training cohort comprised 425 patients treated in 2018 (n = 30), 2019 (n = 97), 2021 (n = 177), and 2022 (n = 112), while the valida-tion cohort consisted of 130 patients treated in 2020. (Fig 2).Our dataset comprises 33 Potential Predictive Factors. Clas-sification of Potential Predictive Factors is shown in Table 1. At the final follow-up,389 patients (71.71%) achieved good outcomes with AOFAS scores above 80 points, while 166 patients (28.29%) had poor outcomes with scores ≤80 points. the mean AOFAS midfoot function score was 84.84 ± 9.17. The average age was 41.41 ± 16.585 years, with 381 males (68.65%) and 174 females (31.35%), yielding a male-to-female ratio of 2.19:1. As shown in Fig 2, patients aged 41–60 with metatarsal fractures were the most common, while those aged 0–20 were less frequent, with a higher proportion of males across all age groups. Blood type B (20.9%, 116 cases), Han ethnicity (97.48%, 541 cases), married (79.82%, 443 cases), farmers (25.41%, 141 cases), and education level of junior high school or below (31.17%, 173 cases) were more prevalent among hospitalized patients. Regarding BMI, patients with normal weight dominated (55.68%, 309 cases). Most hospital admissions occurred in summer (32.97%, 183 cases) and in plains (89.73%, 498 cases). Among the inpatients, 153 (27.57%) were smokers, and 173 (31.17%) were alcohol consumers. In the AO/OTA classification, type B was the most common (46.30%). The ratio of high to low energy injuries was 1.31:1. Preoperative comorbidities included hyper-tension (64 cases), diabetes (27 cases), coronary heart disease (11 cases), respiratory diseases (106 cases), hepato-biliary diseases (27 cases), anemia (18 cases), hypoproteinemia (17 cases), hyperlipidemia (15 cases), hyperuricemia (16 cases), hypocalcemia (9 cases), hypokalemia (12 cases), and previous cerebral infarction (7 cases). Most patients received treatment with steel pins (45.95%, 255 cases) and general anesthesia (56.22%, 312 cases). The average hos-pital stay was 13.12 days, with the majority of patients hospitalized for ≤7 days (47.39%, 263 cases). The vast majority of

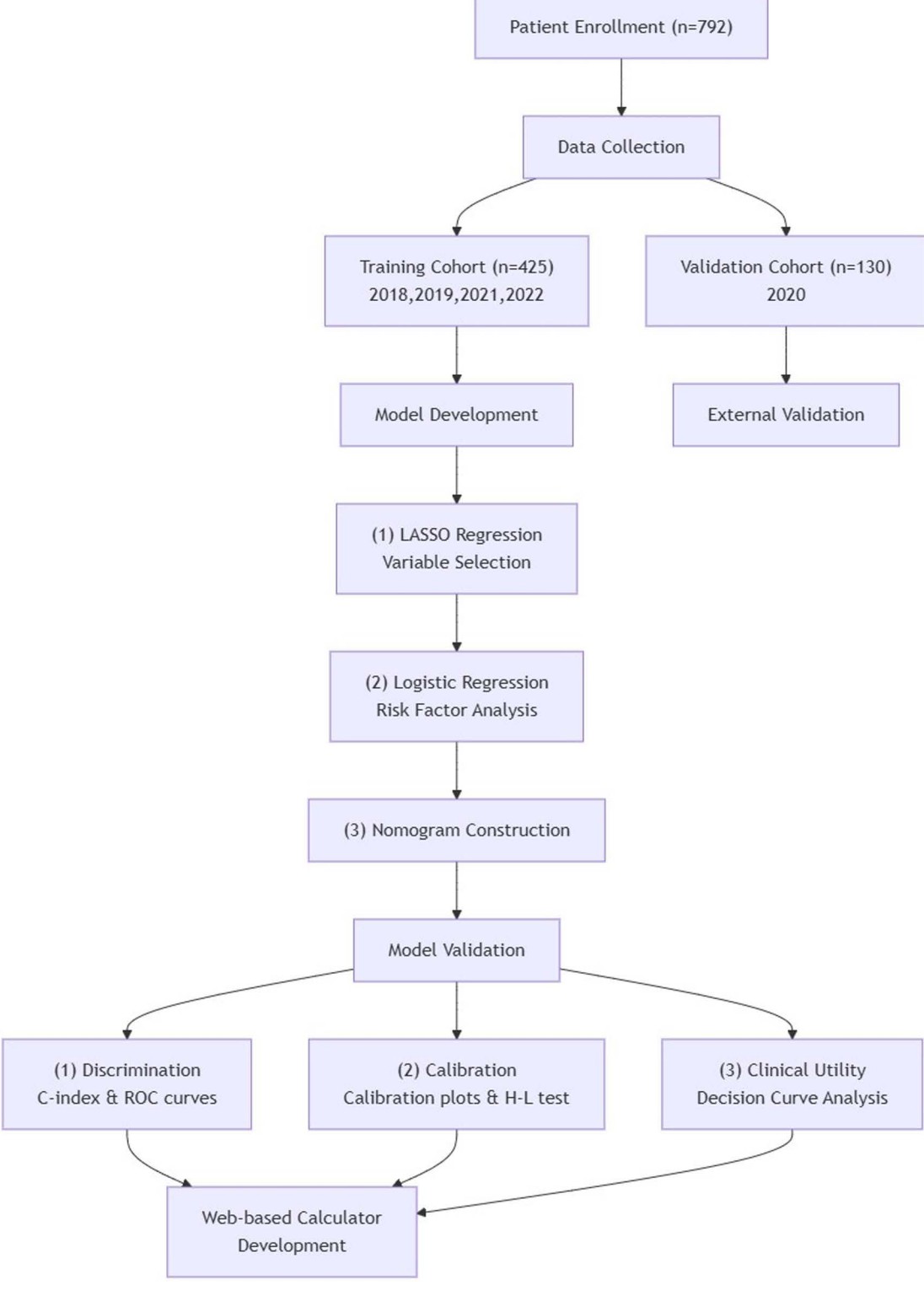

**Fig 1. Flow chart of the total research methodology.**

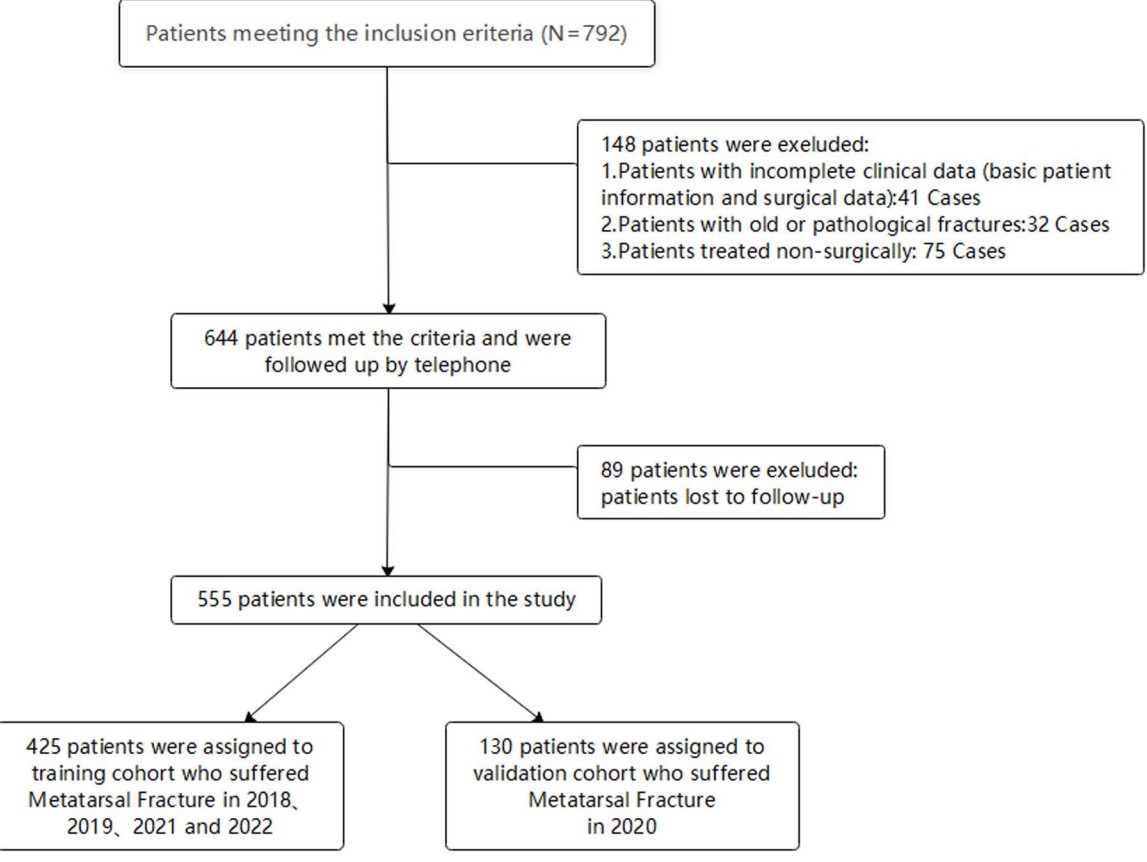

**Fig 2. Flow chart of inclusion and exclusion criteria for metatarsal fracture patients and their division into training and validation sets.**

patients did not receive professional rehabilitation training (83.24%, 462 cases), and 58 patients experienced postoperative infections (10.45%). A comparison of baseline data between the training and validation groups is shown in Table 2.

## Variable selection

Our dataset comprises 33 variables. This dataset has been meticulously verified to include a range of demographic details, clinical cases, and imaging data pre-, intra-, and post-operatively. We utilized LASSO regression technique, performed using the glmnet package in R, combined with a ten-fold cross-validation strategy to determine the optimal regularization parameter (λ). The selection of λ followed a method based on minimizing the mean cross-validation error, which directly corresponds to the lowest prediction error, thereby yielding the best-performing model (Fig 3,Fig 4). Through this rigorous analytical process, we successfully narrowed down the collected factors to 17 key indicators, which were statistically significantly associated with postoperative functional recovery following metatarsal fractures. These key indicators include length of hospital stay, age, education level, ethnicity, occupation, topography, fracture type, injury type, BMI, postoperative infection, rehabilitation training, smoking, blood type, coronary heart disease, anemia, hypocalcemia, and prior cerebral infarction (Table 3). The retained variables displayed non-zero coefficients within the LASSO model and are considered pivotal in revealing the complex interactions between systemic biological mechanisms and postoperative functional recovery following metatarsal fractures. Incorporation into a logistic regression model identified independent risk factors related to postoperative functional recovery, such as age, education level, topography, fracture type, injury type, BMI, postoperative infection, rehabilitation training, smoking, and anemia (Table 4).

**Table 1. Classification of Potential Predictive Factors for Postoperative Functional Recovery in Metatarsal Fracture Patients.**

| Classification | Specific Factors |
| --- | --- |
| Preoperative Factors | Gender |
| | Age |
| | Ethnicity |
| | Occupation |
| | Marital Status |
| | Educational Level |
| | BMI |
| | Blood Type |
| | Season |
| | Terrain |
| | Smoking |
| | Alcohol Consumption |
| | Cause of Injury |
| | Fracture Classification |
| | Common Preoperative Diseases |
| | Diabetes |
| | Hypertension |
| | Coronary Heart Disease |
| | Anemia |
| | Previous Cerebral Infarction |
| | Respiratory Diseases |
| | Hepatobiliary Diseases |
| | Hypoproteinemia |
| | Hyperlipidemia |
| | Hyperuricemia |
| | Hypocalcemia |
| | Hypokalemia |
| Intraoperative Factors | Type of Surgical Fixation |
| | Level of Surgical Procedure |
| | Type of Anesthesia |
| Postoperative Factors | Length of Hospital Stay |
| | Postoperative Infection |
| | Rehabilitation Training |

Residing in hilly areas (OR = 0.183, 95% CI 0.056–0.593), smoking (OR = 0.4, 95% CI 0.204–0.785), obesity (OR = 0.270, 95% CI 0.084–0.869), and undergoing rehabilitation training (OR = 0.237, 95% CI 0.096–0.581) were found to promote postoperative functional recovery. In contrast, low education level (OR = 3.884, 95% CI 1.263–11.948), advanced age (OR = 2.751, 95% CI 1.072–7.061), high-energy injuries (OR = 3.003, 95% CI 1.533–5.885), residing in mountainous areas (OR = 4.671, 95% CI 1.085–20.111), infections (OR = 16.946, 95% CI 0.067–31.848), and anemia (OR = 5.787, 95% CI 1.051–31.861) were detrimental to postoperative functional recovery.

## Model validation and nomogram construction

The C-index for the training cohort was 0.832 (0.7798, 0.8835), and for the validation cohort, it was 0.838 (0.7472, 0.929), indicating a moderate level of discrimination for the model. Receiver Operating Characteristic (ROC) curves for both the

**Table 2. Comparison Analysis of Potential Predictive Factors for Postoperative Functional Recovery in Patients with Metatarsal Fractures between Training and Validation Sets.n (%).**

| Variable | All | Training cohort | Validation cohort | χ2 value | P value |
|---|---|---|---|---|---|
| Metatarsal Fracture Functional Scoring | | | | | |
| good | 398 | 305 | 93 | 0.003 | 0.96 |
| poor | 157 | 120 | 37 | | |
| gender | | | | | |
| male | 381 | 294 | 87 | 0.235 | 0.628 |
| female | 174 | 131 | 43 | | |
| Age | | | | | |
| 0-20 | 64 | 52 | 12 | 5.102 | 0.164 |
| 21-40 | 186 | 149 | 37 | | |
| 41-60 | 218 | 164 | 54 | | |
| >60 | 87 | 60 | 27 | | |
| Educational Background | | | | | |
| Primary and below | 173 | 126 | 47 | 2.665 | 0.466 |
| Middle School Education | 101 | 76 | 25 | | |
| High School Education | 160 | 127 | 33 | | |
| Undergraduate and above | 121 | 96 | 25 | | |
| Ethnic origin | | | | | |
| Han | 541 | 413 | 128 | 0.669 | 0.414 |
| Others | 14 | 12 | 2 | | |
| Occupation | | | | | |
| Manual worker | 91 | 76 | 15 | 8.642 | 0.124 |
| Farmer | 141 | 101 | 40 | | |
| Student | 98 | 74 | 24 | | |
| Office worker | 53 | 45 | 8 | | |
| Retired or Unemployed | 49 | 34 | 15 | | |
| Others | 119 | 95 | 24 | | |
| Topography | | | | | |
| Plain | 498 | 379 | 119 | 1.593 | 0.451 |
| Hill | 34 | 29 | 5 | | |
| Mountainous Area | 23 | 17 | 6 | | |
| Smoking | | | | | |
| Yes | 153 | 119 | 34 | 0.17 | 0.68 |
| No | 402 | 306 | 96 | | |
| Drinking Alcohol | | | | | |
| Yes | 173 | 134 | 39 | 0.109 | 0.742 |
| No | 382 | 291 | 91 | | |
| BMI (kg/m2) | | | | | |
| <18.5 | 13 | 9 | 4 | 0.744 | 0.863 |
| 18.5–23.9 | 309 | 236 | 73 | | |
| 24–27.9 | 160 | 122 | 38 | | |
| ≥ 28.0 | 73 | 58 | 15 | | |
| Marital Status | | | | | |
| Single | 104 | 88 | 16 | 9.052 | 0.029 |
| Married | 443 | 332 | 111 | | |
| Widowed | 3 | 3 | 0 | | |
| Divorced | 5 | 2 | 3 | | |

*(Continued)*

**Table 2.** (Continued)

| Variable | All | Training cohort | Validation cohort | χ2 value | P value |
|---|---|---|---|---|---|
| Blood Type | | | | | |
| A | 58 | 44 | 14 | 5.021 | 0.285 |
| B | 116 | 90 | 26 | | |
| O | 83 | 66 | 17 | | |
| AB | 24 | 14 | 10 | | |
| Not Specified or Not Checked | 277 | 214 | 63 | | |
| Season | | | | | |
| Spring | 137 | 107 | 30 | 0.414 | 0.937 |
| Summer | 183 | 141 | 42 | | |
| Autumn | 163 | 123 | 40 | | |
| Winter | 72 | 54 | 18 | | |
| Fracture Classification | | | | | |
| A | 171 | 133 | 38 | 0.969 | 0.616 |
| B | 257 | 192 | 65 | | |
| C | 127 | 100 | 27 | | |
| Type of Injury | | | | | |
| High-energy injury | 315 | 239 | 76 | 0.201 | 0.654 |
| Low-energy injury | 240 | 186 | 54 | | |
| Diabetes Mellitus | | | | | |
| Yes | 27 | 21 | 6 | 0.023 | 0.88 |
| No | 528 | 404 | 124 | | |
| Hypertension | | | | | |
| Yes | 64 | 52 | 12 | 0.881 | 0.348 |
| No | 491 | 373 | 118 | | |
| Coronary Artery Disease CAD | | | | | |
| Yes | 11 | 7 | 4 | 1.048 | 0.306 |
| No | 544 | 418 | 126 | | |
| Respiratory Diseases | | | | | |
| Yes | 106 | 75 | 31 | 2.476 | 0.116 |
| No | 449 | 350 | 99 | | |
| Hepatobiliary Diseases | | | | | |
| Yes | 27 | 21 | 6 | 0.23 | 0.88 |
| No | 528 | 404 | 124 | | |
| Anemia | | | | | |
| Yes | 18 | 17 | 1 | 3.311 | 0.069 |
| No | 537 | 408 | 129 | | |
| Hypoproteinemia | | | | | |
| Yes | 17 | 13 | 4 | 0.0001 | 0.992 |
| No | 538 | 412 | 126 | | |
| Hyperlipidemia | | | | | |
| Yes | 15 | 10 | 5 | 0.884 | 0.358 |
| No | 540 | 415 | 125 | | |
| Hyperuricemia | | | | | |
| Yes | 16 | 11 | 5 | 0.563 | 0.453 |
| No | 539 | 414 | 125 | | |

*(Continued)*

**Table 2.** (Continued)

| Variable | All | Training cohort | Validation cohort | χ2 value | P value |
|---|---|---|---|---|---|
| Hypocalcemia | | | | | |
| Yes | 9 | 6 | 3 | 0.501 | 0.479 |
| No | 546 | 419 | 127 | | |
| Hypokalemia | | | | | |
| Yes | 12 | 8 | 4 | 0.672 | 0.413 |
| No | 543 | 417 | 126 | | |
| Old Cerebral Infarction | | | | | |
| Yes | 7 | 4 | 3 | 1.493 | 0.222 |
| No | 548 | 421 | 127 | | |
| Length of Hospital Stay | | | | | |
| ≤7 | 263 | 215 | 48 | 8.432 | 0.077 |
| 8-14 | 188 | 135 | 53 | | |
| 15-21 | 47 | 32 | 15 | | |
| 22-28 | 19 | 15 | 4 | | |
| >28 | 38 | 28 | 10 | | |
| Fracture Fixation Method | | | | | |
| Plate Fixation | 171 | 120 | 51 | 10.265 | 0.016* |
| Pin Fixation | 255 | 200 | 55 | | |
| Screw Fixation | 82 | 62 | 20 | | |
| Other | 47 | 43 | 4 | | |
| Type of Anesthesia | | | | | |
| General Anesthesia | 312 | 230 | 82 | 3.246 | 0.072 |
| Local Anesthesia | 243 | 195 | 48 | | |
| Postoperative Infection | | | | | |
| Yes | 58 | 43 | 15 | 0.215 | 0.643 |
| No | 497 | 382 | 115 | | |
| Rehabilitation training | | | | | |
| Yes | 95 | 73 | 22 | 0.019 | 0.892 |
| No | 462 | 352 | 110 | | |
| Sum | 555 | 425 | 130 | | |

*P < 0.05

training and validation cohorts were constructed (Fig 5,Fig 6). Calibration plots (Fig 7,Fig 8,Fig 9) showed that the model's fit curve is close to the ideal diagonal line, suggesting reasonable calibration capability. The Hosmer-Lemeshow (H-L) test demonstrated good model fit, with a P-value of 0.994 for the training cohort and 0.648 for the validation cohort.

As illustrated in Fig 10 and Fig 11, according to the Decision Curve Analysis (DCA), the clinical efficacy is optimal when the threshold probability is in the range of (0.16) 0.72 to 1.00, delivering a higher net benefit of treatment decisions. A nomogram was utilized to visualize the results of the risk prediction model (Fig 12, Fig 13). In practical applications, the risk of poor postoperative functional recovery from metatarsal fracture can be assessed based on individual variables.For individuals who are not anemic, non-smokers, have not undergone rehabilitation training, have no postoperative infections, possess a BMI between 24 and 27.9, have sustained high-energy injuries, have a fracture classified as type C, hold a bachelor's degree or higher, and are over 60 years of age, the corresponding scores can be obtained on the nomogram based on the values of each factor. The likelihood of achieving good functional recovery post-surgery is 0.489 (Fig 13).

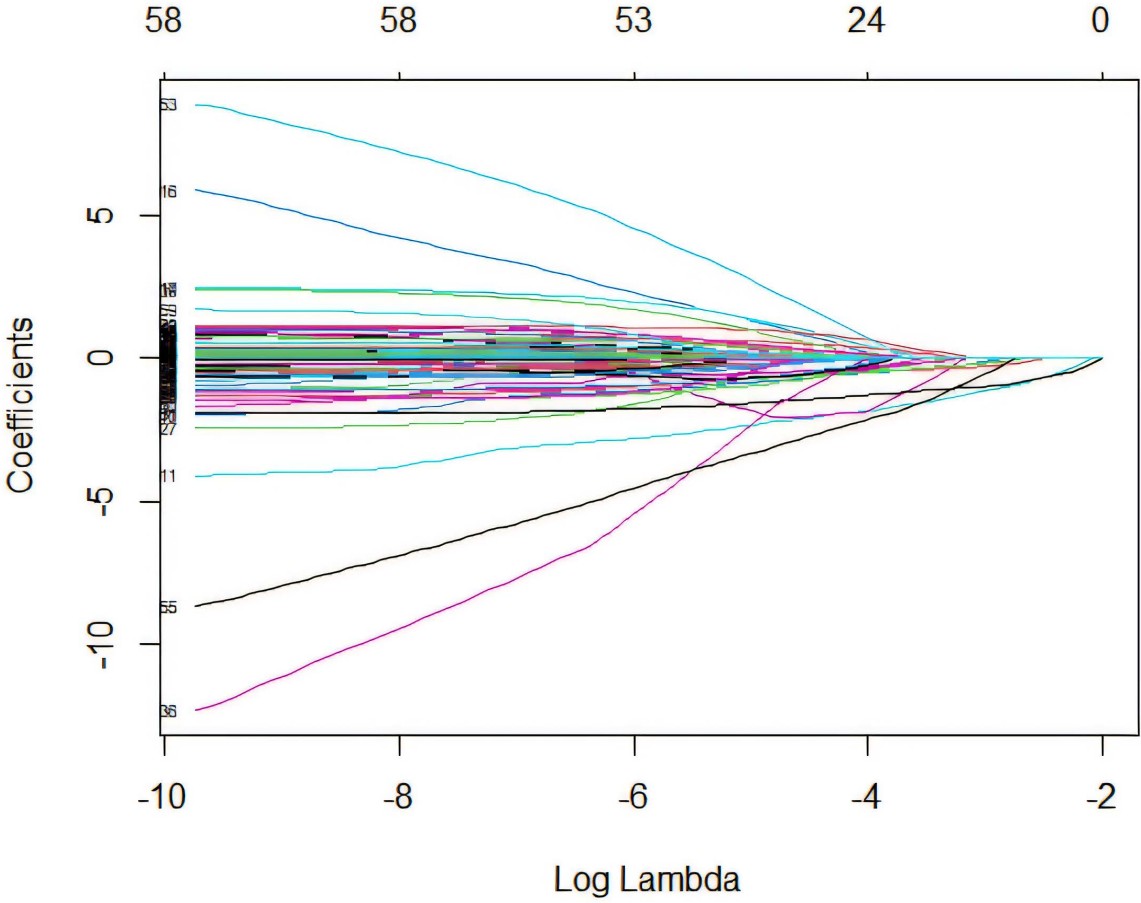

**Fig 3. LASSO regression coefficient profiles: displaying the progression of coefficients of various predictors as the regularization parameter (lambda) is increased.** (Each line represents a different predictor variable in the LASSO regression model.).

An interactive web-based calculator was developed using R Shiny, accessible online at: https://metarecoverypredictor.shinyapps.io/DynNomapp/. By inputting clinical features, one can calculate the probability and 95% confidence interval (CI) for postoperative functional recovery (Fig 14). For instance, for individuals aged 21–40 years with a college degree or higher, a BMI of 18.5–23.9, non-smokers, residing in plains, fracture type A, low-energy injury, no postoperative infection, no rehabilitation training, and non-anemic, selecting "predict" computes the probability of good postoperative recovery at 0.862(95% CI0.709–0.941) (Fig 15). For those aged 41–60 years with primary school education or lower, a BMI of 18.5–23.9, smokers, residing in plains, fracture type B, low-energy injury, postoperative infection, no rehabilitation training, and non-anemic, the probability is 0.706(95% CI0.370–0.908)(Fig 16).

## Discussion

Metatarsal fractures are a common type of bone fracture, accounting for 5–6% of all fractures [4]. These fractures not only affect a patient's ability to walk but may also lead to long-term pain and functional impairment, severely impacting quality of life [45]. Despite the high incidence of metatarsal fractures, comprehensive epidemiological studies on postoperative functional recovery remain limited. This study employs the Least Absolute Shrinkage and Selection Operator (LASSO) regression method to establish and validate, for the first time, a predictive model for postoperative functional recovery in

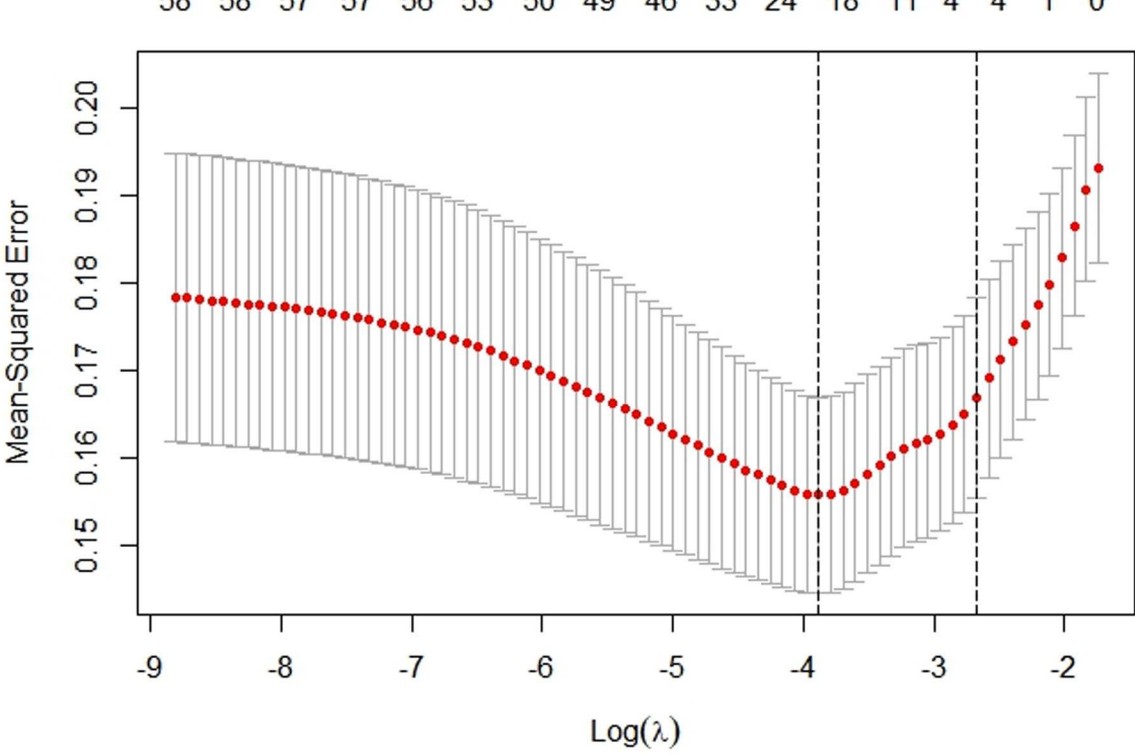

**Fig 4. Selection of Lambda in LASSO Regression: This graph shows the cross-validation curve for model tuning.** (The lambda.min is highlighted, indicating the optimal level of penalization for the LASSO model.).

**Table 3. Coefficients and lambda.min value of the LASSO regression.**

| Variable.Variable | Variable.Coefficient | lambda.min |
|---|---|---|
| Length of Hospital Stay | -0.1153718 | 0.02063499 |
| Age | -1.1354984 | |
| Educational background | -0.3620962 | |
| Ethnic origin | 0.2831308 | |
| Occupation | 0.5591290 | |
| Topography | -0.6193057 | |
| Fracture Classification | -0.5770252 | |
| Type of Injury | -0.6323288 | |
| BMI | -0.5085177 | |
| Postoperative Infection | -1.8185475 | |
| Rehabilitation training | 0.5262817 | |
| Smoking | 0.2662417 | |
| Blood.Type | 0.2204954 | |
| Coronary Artery Disease | -0.7303156 | |
| Anemia | -0.6494542 | |
| Hypocalcemia | 0.6582068 | |
| Old Cerebral Infarction | -0.6981512 | |

**Table 4. Multivariate logistic regression analysis results related to Postoperative Functional Recovery of Metatarsal Fractures.**

| | B | S.E | Wald χ 2 | P | OR | 95% CI |
|---|---|---|---|---|---|---|
| Length of Hospital Stay | | | 2.071 | 0.723 | | |
| 0-7 | -0.556 | 0.598 | 0.864 | 0.353 | 0.574 | 0.178-1.852 |
| 8-14 | -0.52 | 0.558 | 0.867 | 0.352 | 0.595 | 0.199-1.776 |
| 15-21 | -0.676 | 0.699 | 0.935 | 0.334 | 0.509 | 0.129-2.003 |
| >21 | -1.162 | 0.84 | 1.912 | 0.167 | 0.313 | 0.06-1.624 |
| Age | | | 4.75 | 0.191 | | |
| 0-20 | Ref | | | | | |
| 21-40 | 1.211 | 0.991 | 1.492 | 0.222 | 3.356 | 0.481-23.414 |
| 41-60 | 1.012 | 0.481 | 4.43 | 0.035* | 2.751 | 1.072-7.061 |
| >60 | 0.716 | 0.435 | 2.702 | 0.1 | 2.046 | 0.871-4.802 |
| Educational Background | | | 7.099 | 0.069 | | |
| Primary and below | Ref | | | | | |
| Middle School Education | 1.657 | 0.817 | 4.109 | 0.043* | 5.242 | 1.056-26.012 |
| High School Education | 1.357 | 0.573 | 5.602 | 0.018* | 3.884 | 1.263-11.948 |
| Undergraduate and above | 0.606 | 0.476 | 1.62 | 0.203 | 1.833 | 0.721-4.659 |
| Ethnic origin | -1.981 | 1.344 | 2.173 | 0.14 | 0.138 | 0.01-1.921 |
| Occupation | | | 4.203 | 0.521 | | |
| Manual worker | Ref | | | | | |
| Farmer | 0.495 | 0.503 | 0.97 | 0.325 | 1.64 | 0.613-4.392 |
| Student | -0.167 | 0.753 | 0.049 | 0.825 | 0.846 | 0.194-3.7 |
| Office worker | 0.446 | 0.551 | 0.654 | 0.419 | 1.562 | 0.53-4.599 |
| Retired or Unemployed | 0.314 | 0.971 | 0.105 | 0.746 | 1.369 | 0.204-9.174 |
| Others | 1.145 | 0.618 | 3.434 | 0.064 | 3.143 | 0.936-10.553 |
| Topography | | | 8.77 | 0.012 | | |
| Plain | Ref | | | | | |
| Hill | -1.700 | .601 | 8.008 | 0.005* | 0.183 | 0.056-0.593 |
| Mountainous Area | 1.541 | 0.745 | 4.282 | 0.039* | 4.671 | 1.085-20.111 |
| Fracture Classification | | | 10.97 | 0.004 | | |
| A | Ref | | | | | |
| B | 1.156 | 0.396 | 8.521 | 0.004* | 3.179 | 1.462-6.909 |
| C | 1.049 | 0.349 | 9.022 | 0.003* | 2.855 | 1.44-5.662 |
| Type of Injury | 1.1 | 0.343 | 10.267 | 0.001* | 3.003 | 1.533-5.885 |
| BMI (kg/m2) | | | 5.9 | 0.207 | | |
| <18.5 | 2.34 | 2.053 | 1.299 | 0.254 | 10.386 | 0.186-580.949 |
| 18.5–23.9 | 1.244 | 1.549 | 0.645 | 0.422 | 3.47 | 0.167-72.31 |
| 24–27.9 | -1.309 | .597 | 4.816 | 0.028* | 0.270 | 0.084-0.869 |
| ≥ 28.0 | 0.376 | 1.574 | 0.057 | 0.811 | 1.457 | 0.067-31.848 |
| Postoperative Infection | 2.83 | 0.48 | 34.819 | 0.000* | 16.946 | 6.619-43.38 |
| Rehabilitation training | -1.442 | 0.459 | 9.877 | 0.002* | 0.237 | 0.096-0.581 |
| Smoking | -0.916 | 0.344 | 7.089 | 0.008* | 0.4 | 0.204-0.785 |
| Blood Type | | | 3.096 | 0.542 | | |
| A | Ref | | | | | |
| B | -0.346 | 0.471 | 0.539 | 0.463 | 0.707 | 0.281-1.782 |
| O | -0.188 | 0.362 | 0.27 | 0.604 | 0.828 | 0.407-1.685 |
| AB | 0.435 | 0.45 | 0.936 | 0.333 | 1.545 | 0.64-3.733 |

*(Continued)*

**Table 4.** (Continued)

|  | B | S.E | Wald χ 2 | P | OR | 95% CI |
|---|---|---|---|---|---|---|
| Not Specified or Not Checked | 0.727 | 0.841 | 0.748 | 0.387 | 2.069 | 0.398-10.755 |
| Coronary Artery Disease | 0.804 | 0.79 | 1.036 | 0.309 | 2.234 | 0.475-10.506 |
| Anemia | 1.756 | 0.87 | 4.07 | 0.044* | 5.787 | 1.051-31.861 |
| Hypocalcemia | -1.556 | 1.311 | 1.41 | 0.235 | 0.211 | 0.016-2.754 |
| Old Cerebral Infarction | 2.591 | 1.816 | 2.035 | 0.154 | 13.337 | 0.38-468.453 |

*$P < 0.05$

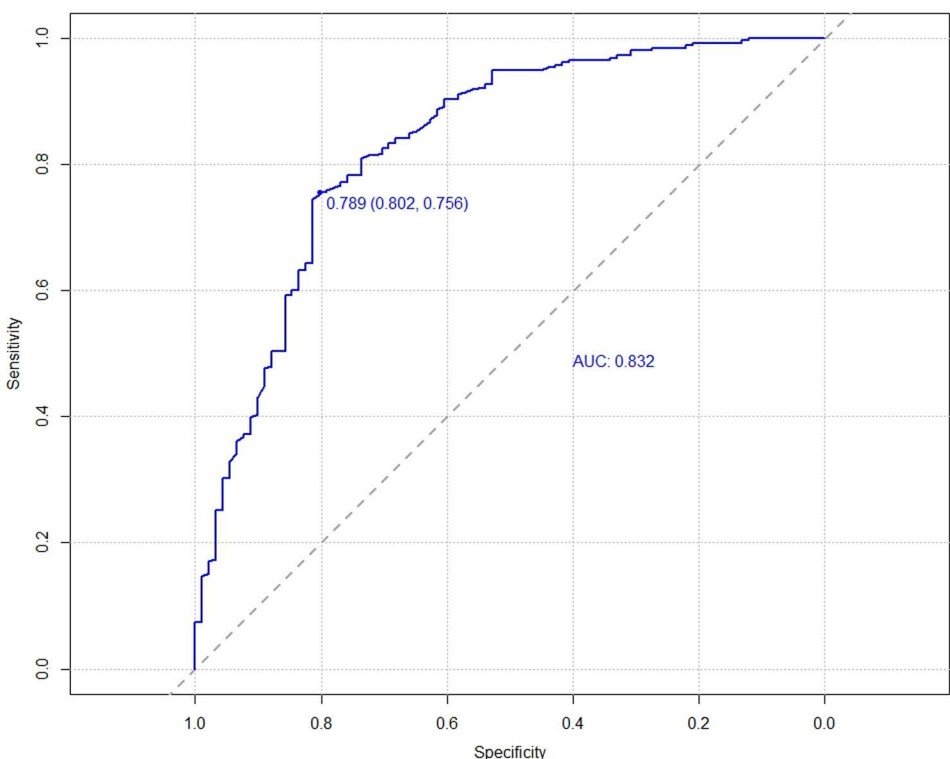

**Fig 5. ROC curve of the prediction model for Postoperative Functional Recovery of Metatarsal Fractures in the Training cohort.**

patients with metatarsal fractures. Furthermore, an interactive calculator based on R Shiny was developed to aid clinicians in devising personalized treatment plans, optimize patient management, and ultimately improve prognosis.

Our findings indicate that obesity, smoking, rehabilitation training, and residing in hilly areas are associated with better postoperative recovery.These results reveal complex interactions between biological, behavioral and environmental factors in fracture healing. The protective effect of obesity on fracture healing appears to operate through multiple mechanisms some studies suggest [46–48] that obese individuals typically have higher bone density. recent studies have revealed that adipose tissue serves as an active endocrine organ. Adipose-derived stem cells (ADSCs) secrete various growth factors and cytokines, such as leptin and specific adipokines, may promote bone repair and angiogenesis [49]. Additionally, the mechanical loading associated with higher body weight may stimulate bone formation through mechanotransduction pathways [50].While our finding that smoking associates with better recovery seems counterintuitive given

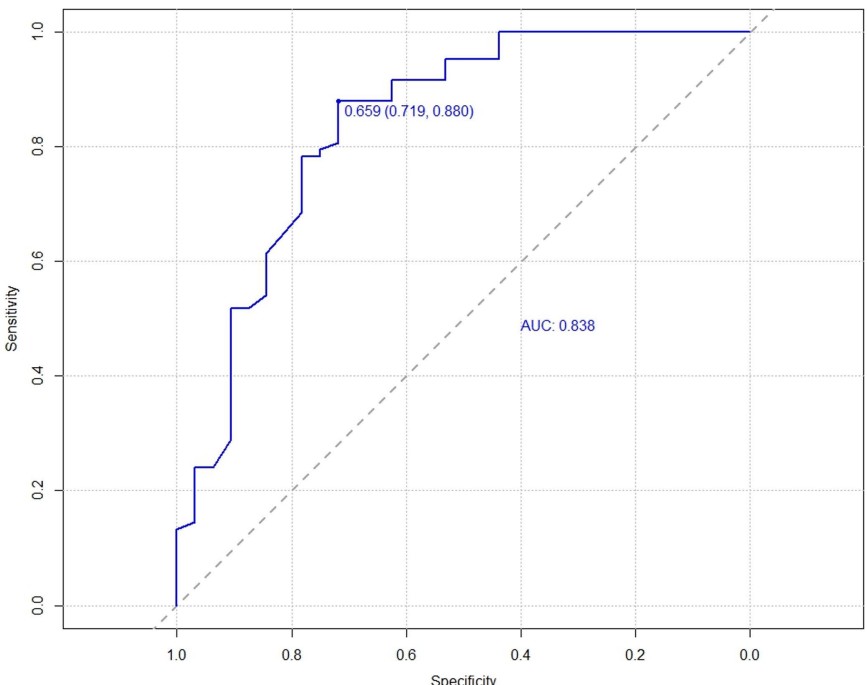

**Fig 6. ROC curve of the prediction model for Postoperative Functional Recovery of Metatarsal Fractures in the Validation cohort.**

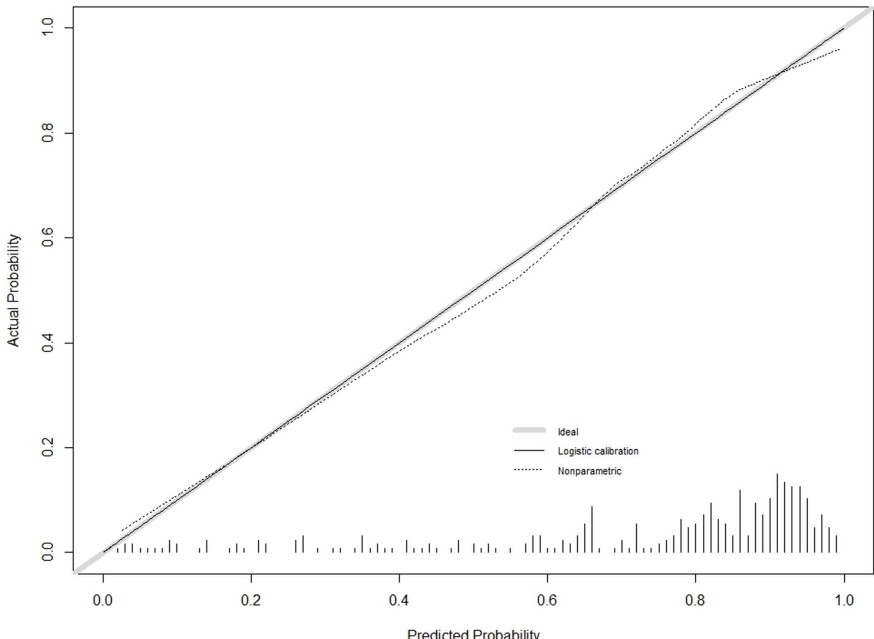

**Fig 7. Calibration curve of the prediction model for Postoperative Functional Recovery of Metatarsal Fractures in the Training cohort.**

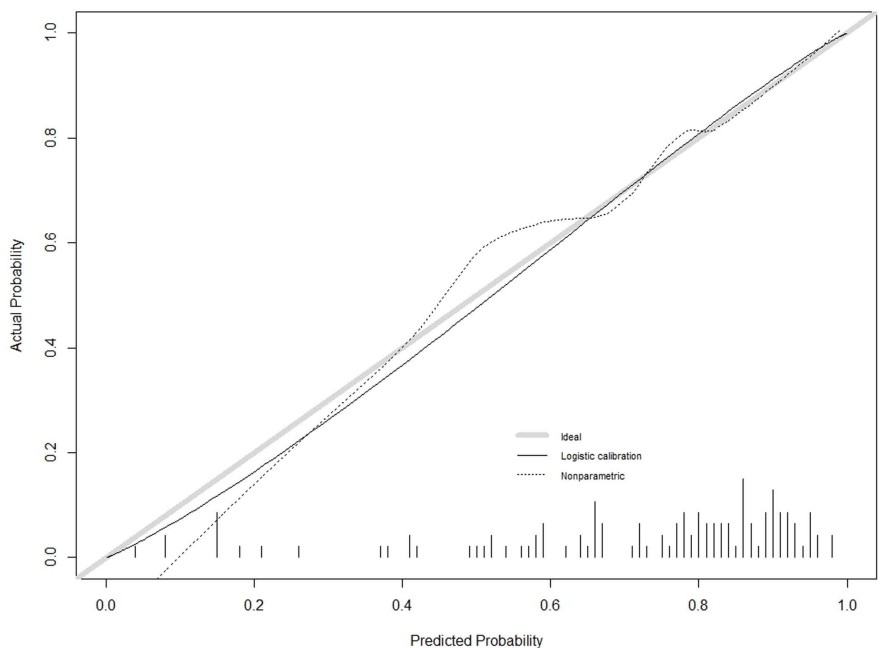

**Fig 8. Calibration curve of the prediction model for Postoperative Functional Recovery of Metatarsal Fractures in the Validation cohort.**

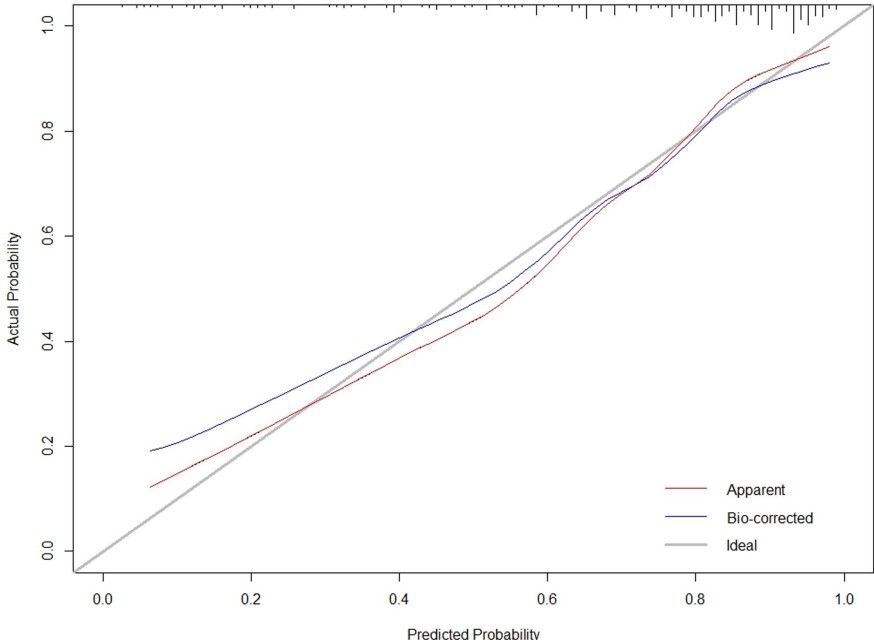

**Fig 9. Calibration curve of the prediction model for Postoperative Functional Recovery of Metatarsal Fractures by internal validation using bootstrap resampling.**

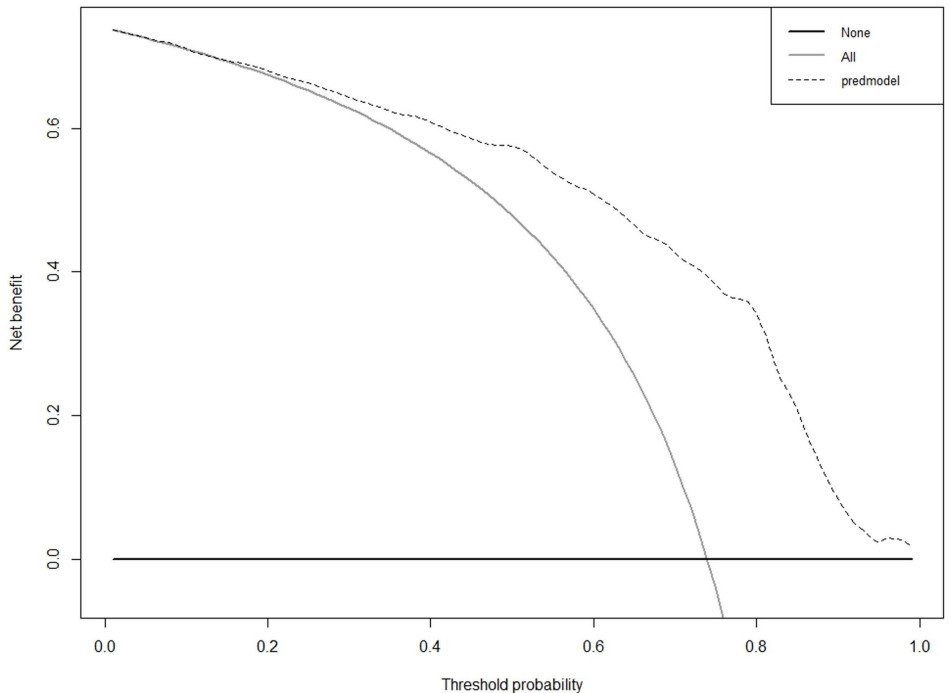

**Fig 10. DCA curve of the prediction model for Postoperative Functional Recovery of Metatarsal Fractures in the Training cohort.**

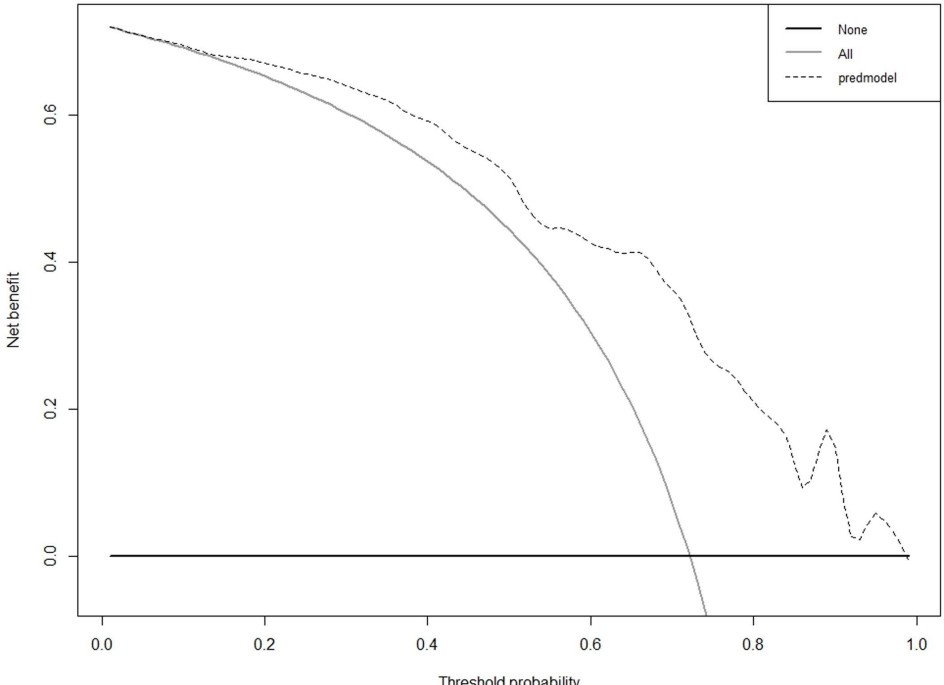

**Fig 11. DCA curve of the prediction model for Postoperative Functional Recovery of Metatarsal Fractures in the Validation cohort.**

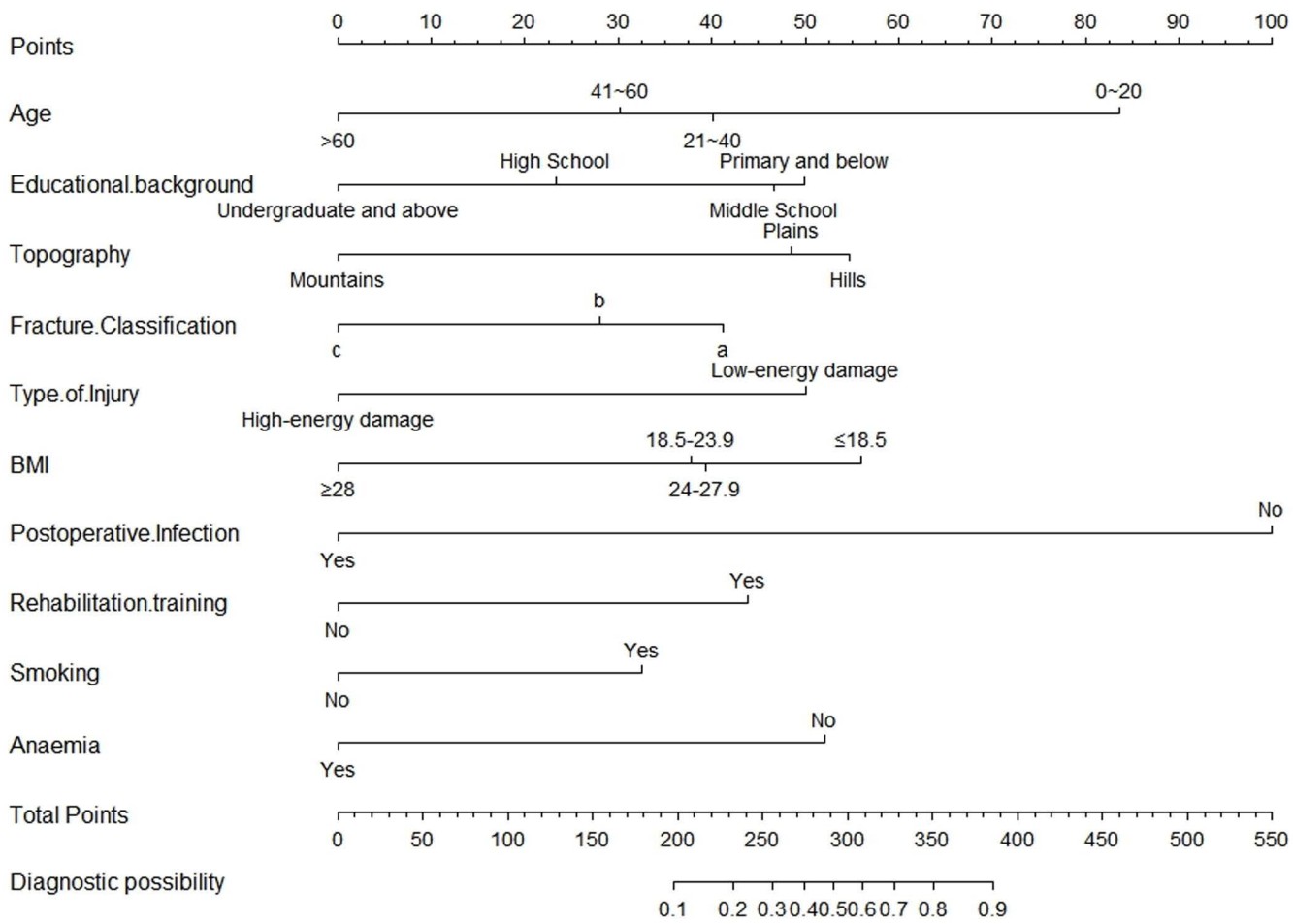

**Fig 12. Nomogram of the prediction model for Postoperative Functional Recovery of Metatarsal Fractures.**

smoking's known negative effects on healing, this may reflect the complex dose-dependent effects of nicotine. Low-dose nicotine has been shown to enhance angiogenesis and increase expression of bone morphogenetic proteins (BMPs) [51], though chronic high-dose exposure remains detrimental. [52,62].Future studies should examine smoking intensity and duration to better understand this relationship. Rehabilitation training has a positive impact on recovery.[53,65],while the source likely extend beyond pure mechanical effects.Recent evidence suggests that appropriate mechanical loading during rehabilitation activates mechanosensitive ion channels in osteocytes, triggering molecular cascades that promote bone remodeling [54]. Furthermore, exercise-induced myokines may enhance bone healing through paracrine signaling [55]. Many studies have [56–58] suggested that living in hilly areas may offer advantages for postoperative functional recovery due to rich natural landscapes, moderate outdoor activity spaces, fresh air, and lower pollution levels, facilitating rehabilitation exercises, enhancing overall health, strengthening immune system function, and promoting fracture healing, which concurs with our study results.

Conversely, low educational attainment, advanced age, high-energy injuries, residing in mountainous areas, infections, and anemia negatively affect postoperative recovery. Clement et al. reported that low educational attainment is associated with poorer health outcomes, possibly due to limited access to healthcare resources and health literacy [59]. Giannoudis et al. noted that advanced age and high-energy injuries are risk factors for delayed fracture healing and complications [60],The

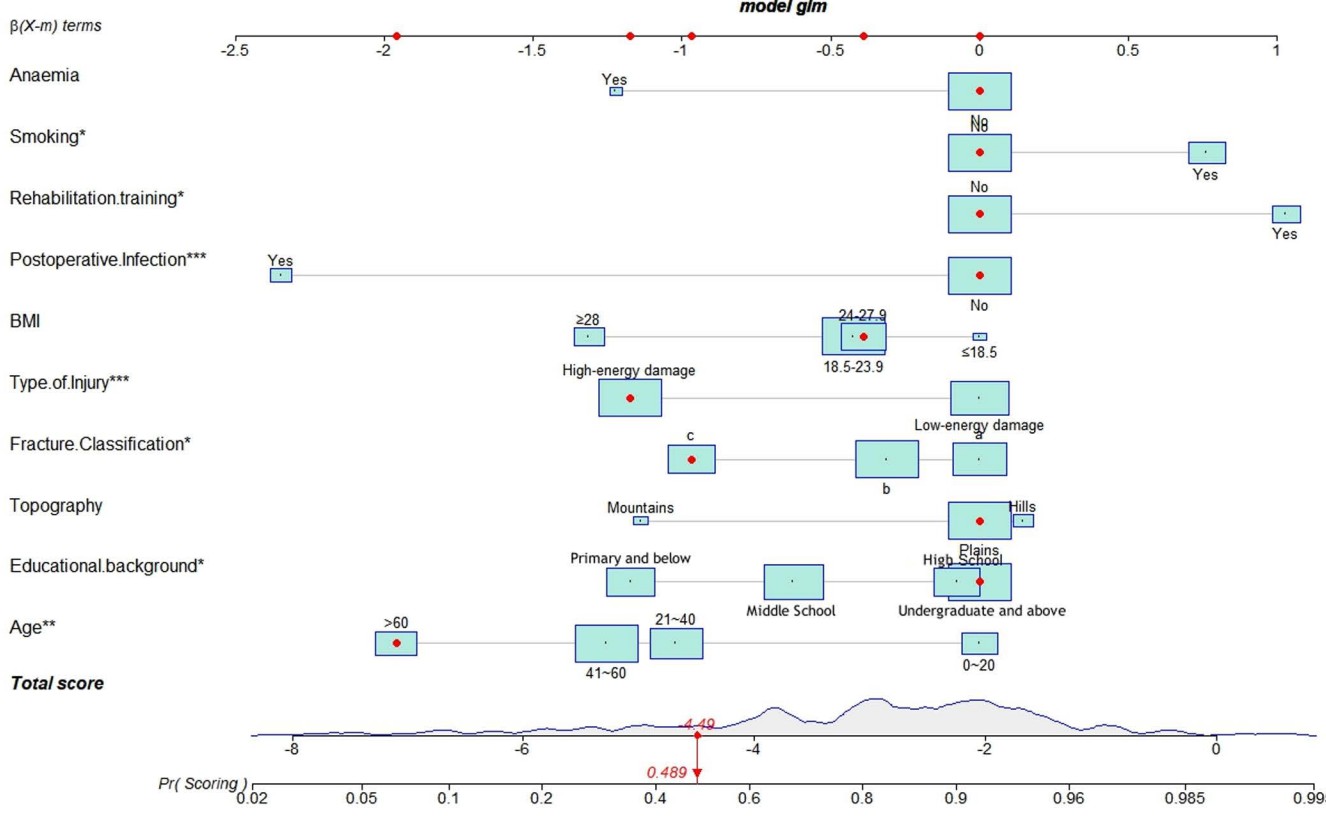

**Fig 13. Schematic diagram of risk scoring on the nomogram.**

negative impact of advanced age on recovery reflects multiple age-related changes: decreased bone mineral density, reduced vascular supply, impaired stem cell function, and altered inflammatory responses [61]. Metsemakers et al. demonstrated that infections undermine tissue repair capacity and negatively affect the healing process [62–65].Anemia's detrimental effect extends beyond simple oxygen transport deficiency,Feng Zhang et al. reported that Deletion of hemoglobin in chondrocytes leads to the loss of Hedy(A membraneless condensate formed by large amounts of hemoglobin), accompanied by severe hypoxia, enhanced glycolysis, and extensive cell death. This indicates an extra-erythrocyte role of hemoglobin in chondrocytes, helping them survive in hypoxic environments [66]. At the same time,Lasocki et al. indicated that anemia leads to insufficient oxygen transport, which affects the metabolic demands of the healing tissues and subsequently significantly influences fracture healing.[63,67] Multiple studies suggest [56,58,68–70] that the uneven terrain of mountainous regions increases postoperative patient inconvenience and fall risk, while often being cold and humid, potentially causing joint and bone discomfort, exacerbating pain, and delaying healing. Furthermore, the scarcity of medical resources can limit postoperative rehabilitation support and professional care, negatively impacting smooth recovery. The differential impact of hilly versus mountainous terrain was an unexpected finding. This paradox may be explained by the 'sweet spot' hypothesis - while moderate terrain variation (hills) provides beneficial exercise opportunities, extreme variation (mountains) creates barriers to mobility and healthcare access. This suggests an optimal level of environmental challenge for recovery.[56,58]

The main strengths of this study include the use of a large sample size and rigorous statistical methods (LASSO regression), enhancing the reliability and accuracy of results. Additionally, a prolonged study period and comprehensive data collection provide a robust foundation for the model's stability. However, the study also has limitations. As a single-center retrospective investigation, selection bias was introduced through the exclusion of patients lost to follow-up

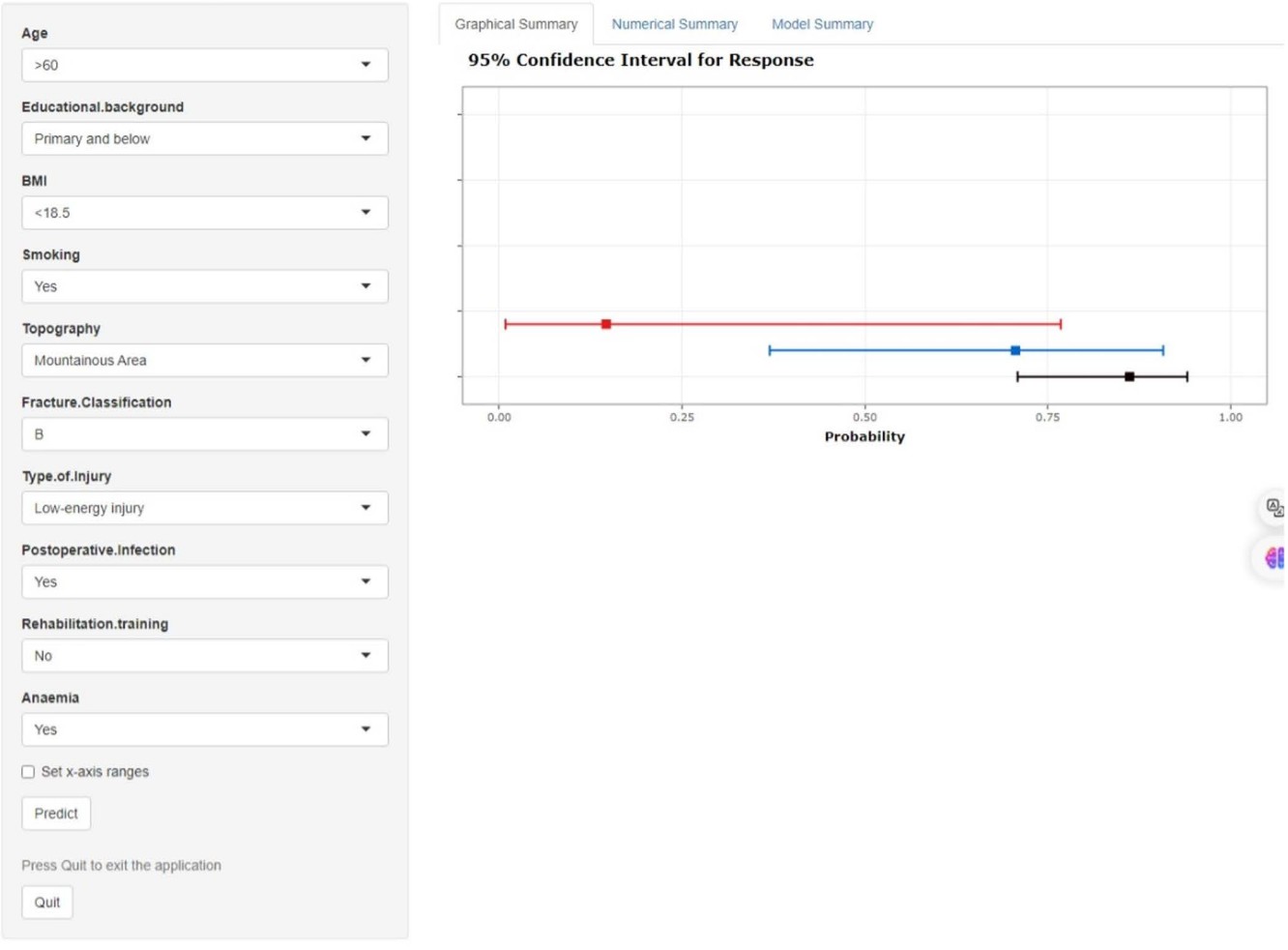

**Fig 14. Webpage illustration of an interactive web-based calculator application using R Shiny.**

and those with incomplete clinical documentation. The study design's exclusive focus on surgically managed cases, omitting conservatively treated patients, potentially limits the comprehensive representation of treatment outcomes. Additionally, the retrospective nature of data collection, relying on medical records and telephone follow-up, introduces potential information bias.The study population's demographic composition presents challenges to external validity, with a predominance of Han ethnicity (97.48%) and male patients (68.65%), confined to a single geographic region in China. Furthermore, the temporal span of data collection (2018–2022) introduces potential confounding variables, including evolving surgical techniques and the impact of the COVID-19 pandemic on the 2020 validation cohort.

Despite the robust findings of our study, several limitations warrant consideration and provide direction for future research and clinical practice. To enhance the generalizability and clinical utility of our findings, future investigations should implement prospective multi-center trials with standardized protocols, incorporating wearable devices for objective rehabilitation monitoring and advanced imaging techniques for precise healing assessment. Additionally, the integration of genetic and molecular markers could elucidate the underlying mechanisms of differential healing responses. Based on our current findings, we recommend implementing comprehensive clinical strategies, including early initiation of supervised

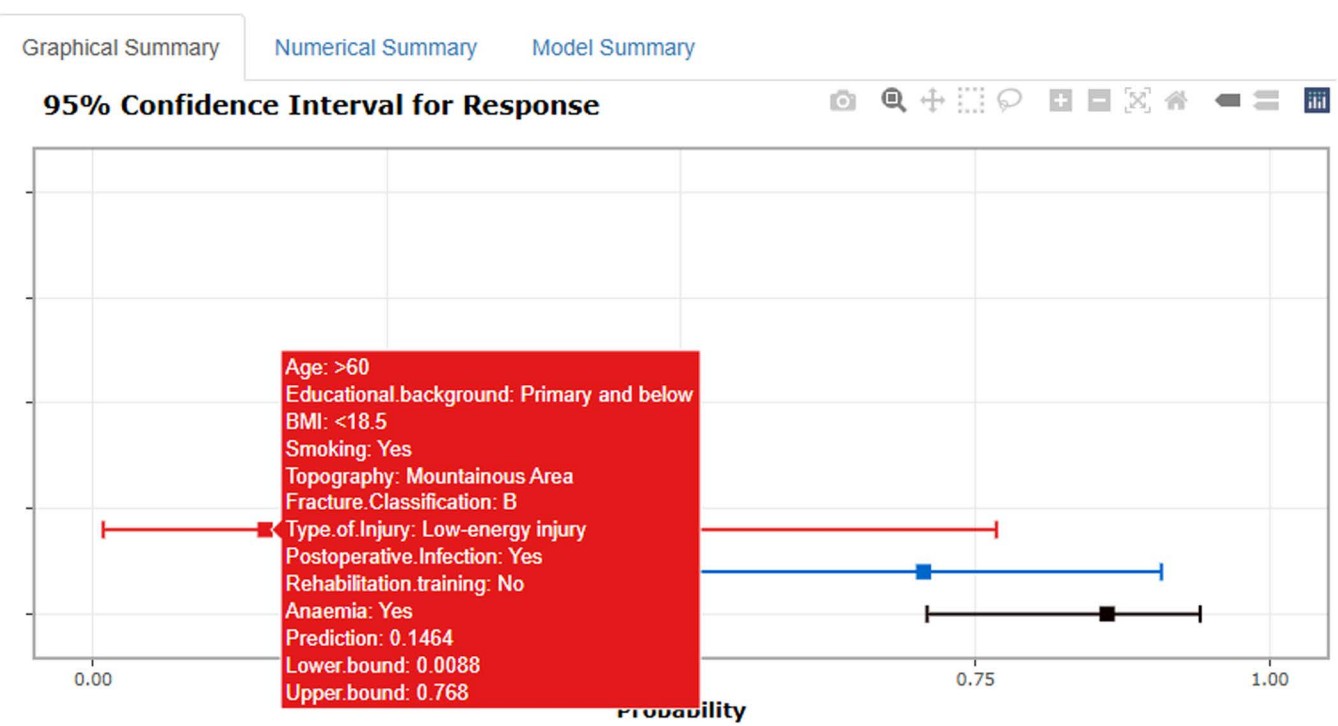

**Fig 15. Example 1 of probability of Postoperative Functional Recovery of Metatarsal Fractures.**

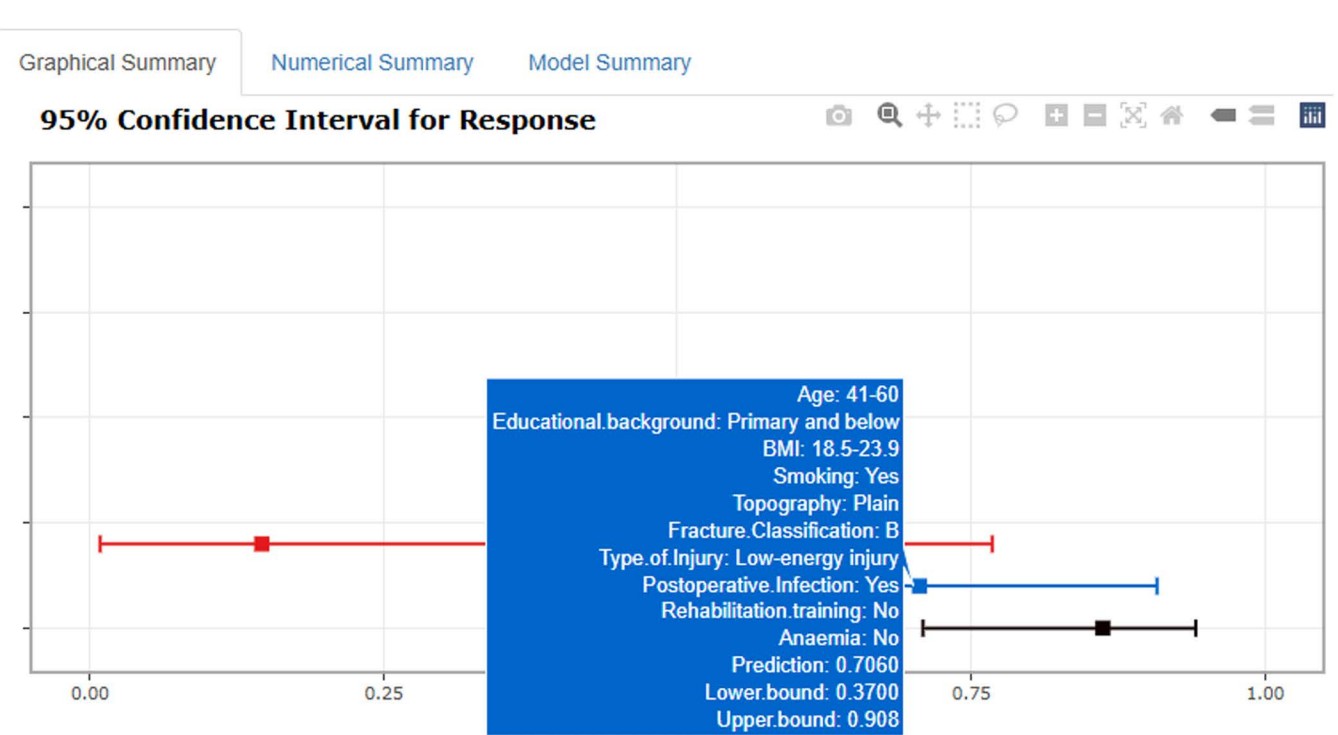

**Fig 16. Example 2 of probability of Postoperative Functional Recovery of Metatarsal Fractures.**

rehabilitation programs, regular monitoring of hemoglobin levels, and targeted interventions for high-risk populations, particularly elderly patients. Furthermore, healthcare providers should consider patients' geographic accessibility when formulating discharge plans, potentially leveraging telemedicine solutions for patients in remote areas. These recommendations, combined with future research addressing the identified limitations, will contribute to optimizing treatment outcomes and advancing our understanding of metatarsal fracture recovery.

## Conclusion

Our results showed that residence in hilly areas, smoking, obesity, and rehabilitation training as being associated with better postoperative recovery, while low educational level, advanced age, high-energy injury, residence in mountainous regions, infection, and anemia were identified as detrimental to postoperative functional recovery. We developed a nomogram and web page to provide personalized convenience for patients and doctors.

## Author contributions

**Formal analysis:** Qian Xiao, Guangzhao Hou, Shuai Zhou, Shihang Liu.

**Funding acquisition:** Wei Chen, Yingze Zhang, Hongzhi Lv.

**Investigation:** Qian Xiao, Guangzhao Hou, Shuai Zhou, Shihang Liu.

**Methodology:** Qian Xiao, Guangzhao Hou, Shuai Zhou, Shihang Liu.

**Resources:** Hongzhi Lv.

**Software:** Qian Xiao, Guangzhao Hou.

**Supervision:** Wei Chen, Yingze Zhang, Hongzhi Lv.

**Visualization:** Wei Chen, Yingze Zhang, Hongzhi Lv.

**Writing – original draft:** Qian Xiao, Guangzhao Hou.

**Writing – review & editing:** Qian Xiao.

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
