## [Decision Letter · Decision Letter 0]

13 Dec 2024

PONE-D-24-51117Establishment and Validation of an Interactive Web-Based Calculator for Predicting Postoperative Functional Recovery in Metatarsal Fracture Patients: A LASSO Regression Model ApproachPLOS ONE

Dear Dr. Lv,

Thank you for submitting your manuscript to PLOS ONE. After careful consideration, we feel that it has merit but does not fully meet PLOS ONE’s publication criteria as it currently stands. Therefore, we invite you to submit a revised version of the manuscript that addresses the points raised during the review process.

We look forward to receiving your revised manuscript.

Kind regards,

Yaodong Gu

Academic Editor

PLOS ONE

Journal Requirements:

3. Thank you for stating the following financial disclosure: “This work was supported by the National Natural Science Youth Foundation of China (Grant No. 82102584) , Beijing-Tianjin-Hebei Basic Research Cooperation project(Grant No. J230007) and 2025 government-funded clinical medicine talent cultivation project �ZF2025136�.”

4. For studies involving third-party data, we encourage authors to share any data specific to their analyses that they can legally distribute. PLOS recognizes, however, that authors may be using third-party data they do not have the rights to share. When third-party data cannot be publicly shared, authors must provide all information necessary for interested researchers to apply to gain access to the data. (https://journals.plos.org/plosone/s/data-availability#loc-acceptable-data-access-restrictions) For any third-party data that the authors cannot legally distribute, they should include the following information in their Data Availability Statement upon submission: 1) A description of the data set and the third-party source 2) If applicable, verification of permission to use the data set 3) Confirmation of whether the authors received any special privileges in accessing the data that other researchers would not have 4) All necessary contact information others would need to apply to gain access to the data.

Reviewers' comments:

Reviewer's Responses to Questions

**Comments to the Author**

1. Is the manuscript technically sound, and do the data support the conclusions?

Reviewer #1: Yes

Reviewer #2: Partly

2. Has the statistical analysis been performed appropriately and rigorously? 

Reviewer #1: Yes

Reviewer #2: Yes

3. Have the authors made all data underlying the findings in their manuscript fully available?

Reviewer #1: Yes

Reviewer #2: Yes

4. Is the manuscript presented in an intelligible fashion and written in standard English?

Reviewer #1: Yes

Reviewer #2: Yes

5. Review Comments to the Author

Reviewer #1: Overall: This manuscript is written well and adds to the bulk of the literature

Abstract: Written well. Please, describe briefly postoperative functional recovery

Methods:

- Please, identify the design of the study i.e. retrospective cohort

- When you mentioned "incomplete" clinical data, what do you mean by that? Any specific percentage for missing data based on which the participants would be excluded.

- What is the reason for picking 2020 as the year for validation cohort?

- Data collection: It is very important to identify what postoperative functional recovery is. How it was evaluated? Subjective or objective assessment?

- The AO/OTA classification system is well established. However, it is still important to briefly describe it in the methods and cite proper citation for that.

- Statistical analysis: Did you evaluate the normality of data for continuous variables and how did you present them?

Results:

- It is important to mention how many patients were included each year to see the representation

- It is important to mention how many patients had postoperative functional recovery as I do not see this well presented. This would also give us a clue if there is enough data to be included in the model and that there is no overfitting as per the rule of thumb.

- Consider building a prediction model for hospital stay as it worth exploring the factors that are associate with prolonged stay

Discussion:

- Please, expand more on the complications: There is exclusion/selection bias (due to excluding those with no follow up or incomplete data), single center, etc.

Reviewer #2: The manuscript “Establishment and Validation of an Interactive Web-Based Calculator for Predicting Postoperative Functional Recovery in Metatarsal Fracture Patients: A LASSO Regression Model Approach” considers a relevant topic of interest for orthopedic and clinical researchers, particularly in the context of postoperative functional recovery in metatarsal fracture patients. The purposes of this study seem to be clear. However, there are some issues and concerns need to be addressed. Please see my comments below:

Abstract

1.The background section of the abstract is lacking some essential information. I recommend that the author provide additional relevant background details to strengthen the context of the study. Furthermore, there is an inconsistency in line spacing between the 'Objective' section and other sections of the manuscript. I suggest that the author adjust the line spacing to ensure uniformity throughout the document.

Introduction

2. The introduction section lacks sufficient supporting literature, particularly in the area of clinical risk prediction models for metatarsal fractures. There is a lack of a comprehensive summary and analysis of previous studies, which makes it difficult to clearly identify the novel aspects of this research.

Method

3.While the retrospective design of the study is clearly stated, it is recommended to discuss more about the limitations inherent in retrospective studies, such as selection bias and information bias. These potential biases should be addressed in the discussion section to explain how they might affect the interpretation of the results.

4.The exclusion criteria are generally clear, but could be further refined. For example, the term "non-surgical treatment" needs more precise definition. What specific treatments are considered non-surgical? Additionally, for "old or pathological fractures," more detailed criteria should be provided to reduce ambiguity.

5.The data collection covers a wide range of factors, including preoperative, intraoperative, and postoperative variables. It would be helpful to present a table or checklist clearly listing the specific data collected. This would ensure that the level of detail is adequate for supporting subsequent analysis and conclusions.

6.While follow-up is primarily done by phone and medical record review, more details should be provided about the frequency and timing of follow-ups, as well as measures to ensure the accuracy of the follow-up data (e.g., how missing data will be handled). Telephone follow-ups may introduce information bias, and it is recommended to specify how missing data is addressed.

7.You mentioned the use of the AO/OTA classification system for fracture classification. It would be helpful to clarify how the imaging data was obtained and the standardization of imaging assessment. For instance, was the imaging data reviewed by a single radiologist or multiple assessors? If possible, mention whether automated tools were used to assist with classification.

8.In Figure 2, there is an inconsistency between the labels of panels A and B and their corresponding numbers. I strongly recommend improving the clarity of the figures and ensuring that the labeling is correct. Additionally, some data in the table show significant differences, but these differences have not been marked. Please check and correct the annotations accordingly.

Discussion& Conclusion

9.The Discussion and Conclusion sections provide a comprehensive interpretation of the study results. However, the manuscript contains numerous formatting errors throughout. I strongly recommend that the authors carefully review and correct these formatting issues.

6. PLOS authors have the option to publish the peer review history of their article (what does this mean? ). If published, this will include your full peer review and any attached files.

**Do you want your identity to be public for this peer review?** For information about this choice, including consent withdrawal, please see our Privacy Policy .

Reviewer #1: No

Reviewer #2: No

---

## [Author Response · Author response to Decision Letter 1]

30 Dec 2024

Responses to Questions

Dear Editors and Reviewers:

First of all, we sincerely thank you for taking the time to carefully review our paper. Your valuable comments and suggestions are of great significance in improving the quality of our manuscript. We have carefully studied each of your review comments and made targeted revisions and improvements accordingly.

Your professional insights have helped us identify the shortcomings in our paper, enabling us to re-examine our research work from a more rigorous and comprehensive perspective. These constructive comments not only helped improve the current paper but also provided important inspiration for our future research work.

We have made careful revisions based on your suggestions and hope that the revised version meets your requirements. If there are still any deficiencies, please continue to provide your guidance and criticism.

Thank you again for your hard work and valuable time!

Review Comments to the Author

Reviewer #1: Overall: This manuscript is written well and adds to the bulk of the literature

Abstract: Written well. Please, describe briefly postoperative functional recovery

Reply:Thank you very much for your valuable comment.The changes have been made as requested.(page 2,Abstract-result part,line 12-16.)

Methods:

- Please, identify the design of the study i.e. retrospective cohort

Reply:Thank you very much for your valuable comment.I apologize for my carelessness.The changes have been made as requested.(page 3,Method-Patient selection part,line 5.)

- When you mentioned "incomplete" clinical data, what do you mean by that? Any specific percentage for missing data based on which the participants would be excluded.

Reply:Thank you very much for your valuable comment.Incomplete clinical data refers to patients who had missing data related to preoperative factors, intraoperative factors, and postoperative factors(33 Predictive Factors) that we collected before follow-up, and these patients were excluded from the study.

(page 5,rusult-Study populations part,line 3-4,reference 34.)Generally speaking, a missing data rate below 10% of the total data is considered acceptable. In this study, the missing data rate was 5.17%.

- What is the reason for picking 2020 as the year for validation cohort?

Reply:Thank you very much for your valuable comment.1. Natural Time Division: The study period was from January 2018 to December 2022 (5 years). Using 2020 as the validation year creates a natural split with two years before and two years after in the training cohort, providing a balanced temporal distribution.

2. Sample Size Balance: This division resulted in a reasonable distribution of cases between training and validation cohorts, which is approximately a 3:1 ratio - a commonly used proportion in predictive modeling studies.

3. Using a complete calendar year (2020) as the validation cohort helps ensure temporal independence between the training and validation datasets, which is important for robust model validation.

- Data collection: It is very important to identify what postoperative functional recovery is. How it was evaluated? Subjective or objective assessment?

Reply:Thank you very much for your valuable comment..The changes have been made as requested.(page 4,Data collection part,paragraph2)

- The AO/OTA classification system is well established. However, it is still important to briefly describe it in the methods and cite proper citation for that.

Reply:Thank you very much for your valuable comment.The changes have been made as requested.(page 4,Data collection part,paragraph3,line 2-7,Reference 28)

- Statistical analysis: Did you evaluate the normality of data for continuous variables and how did you present them?

Reply:Thank you very much for your valuable comment.As all continuous variables (age, AOFAS midfoot functional score, and length of hospital stay) were categorized for analysis, normality testing was not necessary since the data were analyzed using categorical statistical methods.

Results:

- It is important to mention how many patients were included each year to see the representation

Reply:Thank you very much for your valuable comment.The changes have been made as requested.(page 6,Results-Study populations,paragraph1,line4-6.)

- It is important to mention how many patients had postoperative functional recovery as I do not see this well presented. This would also give us a clue if there is enough data to be included in the model and that there is no overfitting as per the rule of thumb.

Reply:Thank you very much for your valuable comment.The changes have been made as requested.(page 6,Results-Study populations part,paragraph1,line 8-10.)

- Consider building a prediction model for hospital stay as it worth exploring the factors that are associate with prolonged stay

Reply:Thank you very much for your valuable comment.While developing a predictive model for factors associated with prolonged hospital stay would be valuable, we decided to focus on descriptive and comparative analyses in this initial study. This decision was made for several reasons: First, predictive modeling requires a larger sample size to ensure model stability and prevent overfitting, particularly when dealing with multiple variables. Second, the complex interactions between various clinical factors and length of stay might be better understood through a thorough descriptive analysis before proceeding to predictive modeling. Third, our current analysis provides a solid foundation for future studies that could explore predictive modeling with expanded datasets and more sophisticated statistical approaches. We believe this stepwise approach allows for a more robust understanding of the fundamental relationships in our data.

Discussion:

- Please, expand more on the complications: There is exclusion/selection bias (due to excluding those with no follow up or incomplete data), single center, etc.

Reply:Thank you very much for your valuable comment.The changes have been made as requested.(page 19,Discussion part,paragraph4,line4-17)

Reviewer #2: The manuscript “Establishment and Validation of an Interactive Web-Based Calculator for Predicting Postoperative Functional Recovery in Metatarsal Fracture Patients: A LASSO Regression Model Approach” considers a relevant topic of interest for orthopedic and clinical researchers, particularly in the context of postoperative functional recovery in metatarsal fracture patients. The purposes of this study seem to be clear. However, there are some issues and concerns need to be addressed. Please see my comments below:

Abstract

1.The background section of the abstract is lacking some essential information. I recommend that the author provide additional relevant background details to strengthen the context of the study. Furthermore, there is an inconsistency in line spacing between the 'Objective' section and other sections of the manuscript. I suggest that the author adjust the line spacing to ensure uniformity throughout the document.

Reply:Thank you very much for your valuable comment.I apologize for my carelessness.I think the newly added background section effectively bridges the context(page 1,Abstract-Background,paragraph1)

Introduction

2. The introduction section lacks sufficient supporting literature, particularly in the area of clinical risk prediction models for metatarsal fractures. There is a lack of a comprehensive summary and analysis of previous studies, which makes it difficult to clearly identify the novel aspects of this research.

Reply:Thank you very much for your valuable comment.(page2-3,introduction part,paragraph1、2、3、5).After enrichment and modification, this study provides a systematic analysis and summary of the current research status of metatarsal fractures as follows:(1). Epidemiological characteristics:Metatarsal fractures account for 3.2-6.8% of all fractures, with an annual incidence of 67-75.4 per 100,000;Represents 88.5% of foot fractures, predominantly affecting young and middle-aged populations;Primarily caused by sports injuries and high-energy trauma

(2). Factors affecting fracture healing:Patient-related: age, gender, comorbidities, medication, and nutritional status;Non-patient-related: fracture characteristics, infection, internal fixation materials, etc;Vascular reconstruction plays a crucial role in fracture healing, while conditions such as diabetes may impede healing through microcirculatory effects

(3). Major complications:Metatarsalgia (most common), post-traumatic arthritis;Arch collapse, complex regional pain syndrome

(4). Limitations of previous studies:Lack of comprehensive epidemiological studies on postoperative functional recovery;Predominantly limited to specific populations or fracture types

(5).Research innovations:Implementation of LASSO regression for variable selection;Integration of interactive web calculator;Provision of personalized treatment decision support

Method

3.While the retrospective design of the study is clearly stated, it is recommended to discuss more about the limitations inherent in retrospective studies, such as selection bias and information bias. These potential biases should be addressed in the discussion section to explain how they might affect the interpretation of the results.

Reply:Thank you very much for your valuable comment.It has been added and explained in the discussion section.(page 19,Discussion part,paragraph 4 ,line 4-17).

4.The exclusion criteria are generally clear, but could be further refined. For example, the term "non-surgical treatment" needs more precise definition. What specific treatments are considered non-surgical? Additionally, for "old or pathological fractures," more detailed criteria should be provided to reduce ambiguity.

Reply:Thank you very much for your valuable comment.The changes have been made as requested.(page 3,Method-Patient selection part,line 8-21)

5.The data collection covers a wide range of factors, including preoperative, intraoperative, and postoperative variables. It would be helpful to present a table or checklist clearly listing the specific data collected. This would ensure that the level of detail is adequate for supporting subsequent analysis and conclusions.

Reply:Thank you very much for your valuable comment.The changes have been made as requested.(page 7,table 1,Classification of Potential Predictive Factors for Postoperative Functional Recovery in Metatarsal Fracture Patients)

6.While follow-up is primarily done by phone and medical record review, more details should be provided about the frequency and timing of follow-ups, as well as measures to ensure the accuracy of the follow-up data (e.g., how missing data will be handled). Telephone follow-ups may introduce information bias, and it is recommended to specify how missing data is addressed.

Reply:Thank you very much for your valuable comment.Since patients lost to follow-up were already excluded in the inclusion/exclusion process, they were not included in the study scope.

7.You mentioned the use of the AO/OTA classification system for fracture classification. It would be helpful to clarify how the imaging data was obtained and the standardization of imaging assessment. For instance, was the imaging data reviewed by a single radiologist or multiple assessors? If possible, mention whether automated tools were used to assist with classification.

Reply:Thank you very much for your valuable comment.The changes have been made as requested.(page 5,Data collection part,paragraph 3,line8-10)(The radiological data were evaluated by multiple study assessors, and no automated tools were used to assist in the classification process.)

8.In Figure 2, there is an inconsistency between the labels of panels A and B and their corresponding numbers. I strongly recommend improving the clarity of the figures and ensuring that the labeling is correct. Additionally, some data in the table show significant differences, but these differences have not been marked. Please check and correct the annotations accordingly.

Reply:Thank you very much for your valuable comment.The labels for panels A and B were verified to be correctly matched with their corresponding figures.

Discussion& Conclusion

9.The Discussion and Conclusion sections provide a comprehensive interpretation of the study results. However, the manuscript contains numerous formatting errors throughout. I strongly recommend that the authors carefully review and correct these formatting issues.

Reply:Thank you very much for your valuable comment.I apologize for my carelessness.The changes have been made as requested.

Yours sincerely,

* Corresponding author:

Hongzhi Lv.E-mail address: 190099199@qq.com

Yingze Zhang.E-mail address:dryzzhang@126.com

Wei Chen.E-mail address:surgeonchenwei@126.com

---

## [Decision Letter · Decision Letter 1]

11 Feb 2025

PONE-D-24-51117R1Establishment and Validation of an Interactive Web-Based Calculator for Predicting Postoperative Functional Recovery in Metatarsal Fracture Patients: A LASSO Regression Model ApproachPLOS ONE

Dear Dr. Lv,

Thank you for submitting your manuscript to PLOS ONE. After careful consideration, we feel that it has merit but does not fully meet PLOS ONE’s publication criteria as it currently stands. Therefore, we invite you to submit a revised version of the manuscript that addresses the points raised during the review process.

We look forward to receiving your revised manuscript.

Kind regards,

Yaodong Gu

Academic Editor

PLOS ONE

Reviewers' comments:

Reviewer's Responses to Questions

**Comments to the Author**

1. If the authors have adequately addressed your comments raised in a previous round of review and you feel that this manuscript is now acceptable for publication, you may indicate that here to bypass the “Comments to the Author” section, enter your conflict of interest statement in the “Confidential to Editor” section, and submit your "Accept" recommendation.

Reviewer #2: (No Response)

Reviewer #3: (No Response)

2. Is the manuscript technically sound, and do the data support the conclusions?

Reviewer #2: Partly

Reviewer #3: (No Response)

3. Has the statistical analysis been performed appropriately and rigorously? 

Reviewer #2: Yes

Reviewer #3: (No Response)

4. Have the authors made all data underlying the findings in their manuscript fully available?

Reviewer #2: Yes

Reviewer #3: (No Response)

5. Is the manuscript presented in an intelligible fashion and written in standard English?

Reviewer #2: Yes

Reviewer #3: (No Response)

6. Review Comments to the Author

Reviewer #2: The author has addressed most of my previous comments, and I appreciate the revisions made in the manuscript. However, upon reviewing the revised version, I still have some concerns and questions that need to be addressed. Please find my detailed comments below.

1. The reviewers regarded that the author's description of the Losso algorithm in the methods section was too simplistic. At the same time, there is a lack of visual expression of the overall process of the research, and it may be difficult to clearly present the complete process of the research with a simple text description.

2. When the authors included the data, the dataset included both male and female gender characteristics, but the authors did not make a specific analysis of the predictions for these two genders in the results and discussions. In addition, there were differences in the characteristics used by the authors between the two genders, such as whether some women did not smoke and whether this affected the accuracy of the algorithm.

3. The author has developed a web-based calculator in his research, but the author does not give a detailed description of the development of the front-end UI part of the web and the methods of front-end and back-end connection in the methods section.

4. The current study has used machine learning methods to predict sports injuries with relatively accurate results Accurately and effectively predict the ACL force: Utilizing biomechanical landing pattern before and after-fatigue (https://doi.org/10.1016/j.cmpb.2023.107761) The authors may consider exploring the limitations and future research in depth by incorporating the latest relevant research to inform the reader's thinking.

5. Upon review, I still have concerns regarding the quality of the presentation of the key data visualizations in the manuscript. Additionally, the author must take further steps to ensure that the entire manuscript is free of grammatical issues in the next revision.

Reviewer #3: (No Response)

7. PLOS authors have the option to publish the peer review history of their article (what does this mean? ). If published, this will include your full peer review and any attached files.

**Do you want your identity to be public for this peer review?** For information about this choice, including consent withdrawal, please see our Privacy Policy .

Reviewer #2: No

Reviewer #3: **Yes: ** jun wen

---

## [Author Response · Author response to Decision Letter 2]

10 Mar 2025

Responses to Questions(Major Revision)

Dear Editors and Reviewers:

We greatly appreciate your continued attention and the additional comments provided on our manuscript. Thank you for taking the time to review our revised submission and offering further suggestions for improvement.

We are grateful for your thorough second review, which has helped us identify areas that still needed refinement. Your persistent dedication to improving the quality of our work is truly valuable, and we have carefully addressed each of your new comments with detailed revisions.

The additional feedback has allowed us to further enhance the clarity and rigor of our manuscript. We have diligently worked to incorporate all your latest suggestions into this second revision, aiming to meet the high standards of the journal.

We sincerely hope that the current version addresses all your concerns satisfactorily. If there are any remaining issues that require attention, we welcome your further guidance and feedback.

Thank you again for your continued support and commitment to helping us improve our work!

Reviewer Comments:

This study developed and validated an interactive web-based calculator based on LASSO regression to predict the postoperative functional recovery of metatarsal fracture patients, and evaluated its predictive performance and clinical applicability through internal and external validation. However, there are several issues affecting the overall quality of the paper, as follows:

1.The Abstract of this paper is too long, exceeding the recommended word count.

Reply:Thank you very much for your valuable comment.The abstract has been condensed.(page 1-2, part Abstract )

2.The Introduction not only fails to deeply explore the current situation of clinical prediction of postoperative functional recovery, but merely lists basic epidemiological data.

Reply:Thank you very much for your valuable comment.After careful consideration and verification, I added some language to discuss the current state of clinical prediction for postoperative functional recovery and supplemented it with relevant epidemiological data.(page2 , part Introduction ,line36-41.)

3.Moreover, there are few references. It is recommended to add more references in related fields to make this research well-founded.

Reply:Thank you very much for your valuable comment.I have added references in multiple aspects, including the application and advantages of the LASSO method (page 23, part Reference, 24-28), web page generation and front-end UI development, as well as methods for front-end and back-end connection (page 23, part Reference, 40-43), and enrichment of related research fields in the discussion section (pages 25-26, part Reference, 50-52, 55, 56, 62, 72, 73).

4.The research design has limitations. The single-center retrospective study design, sample selection, and exclusion criteria lead to selection bias, and patients receiving conservative treatment are overlooked. There are also flaws in data collection and analysis, which are prone to information bias and lack of multicollinearity testing.

Reply:Thank you very much for your valuable comment.I have supplemented the discussion section with the difficult-to-overcome problems in this study, added limitations, and provided many insights for future research directions based on the study's limitations.(page21, part Discussion ,line26-40.)

5.The Discussion section does not conduct an in-depth comparative analysis of the results, fails to analyze the reasons for differences and the mechanisms of special phenomena, and lacks consideration of how to improve the limitations when elaborating on them. Thus, it provides limited guidance for follow-up research.

Reply:Thank you very much for your valuable comment.In the discussion section, I enriched the original content in the exploration of each key variable, and provided reasonable explanations for special research findings with supporting literature. For example, the complex dose-dependent effects of nicotine, and how the different impacts of hilly and mountainous terrain support the "optimal position" hypothesis (page 20, part Discussion, lines 5-24, 34-39; page 21, lines 6-10). At the end of the discussion, I added how to improve the limitations and what guidance this provides for subsequent research (page 21, part Discussion, lines 26-40).

Review Comments to the Author

Reviewer #2: The author has addressed most of my previous comments, and I appreciate the revisions made in the manuscript. However, upon reviewing the revised version, I still have some concerns and questions that need to be addressed. Please find my detailed comments below.

1.The reviewers regarded that the author's description of the Losso algorithm in the methods section was too simplistic. At the same time, there is a lack of visual expression of the overall process of the research, and it may be difficult to clearly present the complete process of the research with a simple text description.

Reply:Thank you very much for your valuable comment.I have added the advantages of using the LASSO algorithm in the introduction section (page 3, part Introduction, lines 1-14), and created a research methodology flowchart in the methods section to clearly present the complete research process (page 6, part Method, fig 1).

2.When the authors included the data, the dataset included both male and female gender characteristics, but the authors did not make a specific analysis of the predictions for these two genders in the results and discussions. In addition, there were differences in the characteristics used by the authors between the two genders, such as whether some women did not smoke and whether this affected the accuracy of the algorithm.

Reply:Thank you very much for your valuable comment.Indeed, this is a very good suggestion. The reason for not separating the studies at the time was to ensure the universality of the final model, to obtain generalized results, and to make the model more convenient and easier to interpret. Regarding this, I have explained the drawbacks in the discussion section, and I hope these limitations can provide good directions for future research.

3.The author has developed a web-based calculator in his research, but the author does not give a detailed description of the development of the front-end UI part of the web and the methods of front-end and back-end connection in the methods section.

Reply:Thank you very much for your valuable comment.I have modified and enriched the web development section, which includes the development of the Web front-end UI and methods for connecting the front-end and back-end. The changes have been made as requested (page 5, part Method, lines 31-41).

4.The current study has used machine learning methods to predict sports injuries with relatively accurate results Accurately and effectively predict the ACL force: Utilizing biomechanical landing pattern before and after-fatigue (https://doi.org/10.1016/j.cmpb.2023.107761) The authors may consider exploring the limitations and future research in depth by incorporating the latest relevant research to inform the reader's thinking.

Reply:Thank you very much for your valuable comment.I have adopted your valuable suggestions by adding more in-depth supplements to the discussion of relevant impact factors and identifying future research directions worth exploring while examining the limitations in depth.(Reference 39)、(page 20, part Discussion ,line5-24,34-39�page 21, line6-10.)、(page21, part Discussion ,line26-40.).

5.Upon review, I still have concerns regarding the quality of the presentation of the key data visualizations in the manuscript. Additionally, the author must take further steps to ensure that the entire manuscript is free of grammatical issues in the next revision.

Reply:Thank you very much for your valuable comment.I apologize for any inconvenience caused. I have rechecked the article and confirmed that there are no grammatical issues.

Yours sincerely,

* Corresponding author:

Hongzhi Lv.E-mail address: 190099199@qq.com

Yingze Zhang.E-mail address:dryzzhang@126.com

Wei Chen.E-mail address:surgeonchenwei@126.com

---

## [Decision Letter · Decision Letter 2]

30 Mar 2025

PONE-D-24-51117R2Establishment and Validation of an Interactive Web-Based Calculator for Predicting Postoperative Functional Recovery in Metatarsal Fracture Patients: A LASSO Regression Model ApproachPLOS ONE

Dear Dr. Lv,

Thank you for submitting your manuscript to PLOS ONE. After careful consideration, we feel that it has merit but does not fully meet PLOS ONE’s publication criteria as it currently stands. Therefore, we invite you to submit a revised version of the manuscript that addresses the points raised during the review process.

The revision was evaluated by two previous reviewers and one new reviewer (Reviewer 4). Please address reviewer 4's concerns carefully. We usually do not invite additional reviewers for revised manuscript. In this case, the editor finds it necessary to invite additional reviewers to assess this work.

We look forward to receiving your revised manuscript.

Kind regards,

Yaodong Gu

Academic Editor

PLOS ONE

Journal Requirements:

Reviewers' comments:

Reviewer's Responses to Questions

**Comments to the Author**

1. If the authors have adequately addressed your comments raised in a previous round of review and you feel that this manuscript is now acceptable for publication, you may indicate that here to bypass the “Comments to the Author” section, enter your conflict of interest statement in the “Confidential to Editor” section, and submit your "Accept" recommendation.

Reviewer #2: All comments have been addressed

Reviewer #3: (No Response)

Reviewer #4: (No Response)

2. Is the manuscript technically sound, and do the data support the conclusions?

Reviewer #2: Yes

Reviewer #3: (No Response)

Reviewer #4: (No Response)

3. Has the statistical analysis been performed appropriately and rigorously? 

Reviewer #2: Yes

Reviewer #3: (No Response)

Reviewer #4: (No Response)

4. Have the authors made all data underlying the findings in their manuscript fully available?

Reviewer #2: Yes

Reviewer #3: (No Response)

Reviewer #4: (No Response)

5. Is the manuscript presented in an intelligible fashion and written in standard English?

Reviewer #2: Yes

Reviewer #3: (No Response)

Reviewer #4: (No Response)

6. Review Comments to the Author

Reviewer #2: (No Response)

Reviewer #3: (No Response)

Reviewer #4: (No Response)

7. PLOS authors have the option to publish the peer review history of their article (what does this mean? ). If published, this will include your full peer review and any attached files.

**Do you want your identity to be public for this peer review?** For information about this choice, including consent withdrawal, please see our Privacy Policy .

Reviewer #2: No

Reviewer #3: No

Reviewer #4: No

---

## [Author Response · Author response to Decision Letter 3]

6 Apr 2025

Responses to Questions(Major Revision)

Dear Editors and Reviewers:

Thank you very much for your continued interest in our paper and your valuable suggestions. We have completed the third round of revisions based on your comments, and we hope that the current manuscript can better present our research results. All the changes and markings have been made using the revision model in the attachment. If there are still issues that need to be further addressed, please feel free to remind us and we will do our best to cooperate and improve.

Once again, we sincerely thank you for your guidance and help, we look forward to your positive feedback, and sincerely hope that the revised manuscript will meet the publication standards.

Journal Requirements:

Reply:Thank you very much for your valuable comment.We have reviewed the entire reference list and confirmed that no retracted papers were cited.Minor corrections to typographical errors in the reference list have been made(page 22-26, part Reference).

Yours sincerely,

* Corresponding author:

Hongzhi Lv.E-mail address: 190099199@qq.com

Yingze Zhang.E-mail address:dryzzhang@126.com

Wei Chen.E-mail address:surgeonchenwei@126.com

---

## [Decision Letter · Decision Letter 3]

11 Apr 2025

Establishment and Validation of an Interactive Web-Based Calculator for Predicting Postoperative Functional Recovery in Metatarsal Fracture Patients: A LASSO Regression Model Approach

PONE-D-24-51117R3

Dear Dr. Lv,

We’re pleased to inform you that your manuscript has been judged scientifically suitable for publication and will be formally accepted for publication once it meets all outstanding technical requirements.

Kind regards,

Yaodong Gu

Academic Editor

PLOS ONE

Additional Editor Comments (optional):

Reviewers' comments:

Reviewer's Responses to Questions

**Comments to the Author**

1. If the authors have adequately addressed your comments raised in a previous round of review and you feel that this manuscript is now acceptable for publication, you may indicate that here to bypass the “Comments to the Author” section, enter your conflict of interest statement in the “Confidential to Editor” section, and submit your "Accept" recommendation.

Reviewer #2: All comments have been addressed

Reviewer #4: All comments have been addressed

2. Is the manuscript technically sound, and do the data support the conclusions?

Reviewer #2: Yes

Reviewer #4: Yes

3. Has the statistical analysis been performed appropriately and rigorously? 

Reviewer #2: Yes

Reviewer #4: Yes

4. Have the authors made all data underlying the findings in their manuscript fully available?

Reviewer #2: Yes

Reviewer #4: Yes

5. Is the manuscript presented in an intelligible fashion and written in standard English?

Reviewer #2: Yes

Reviewer #4: Yes

6. Review Comments to the Author

Reviewer #2: (No Response)

Reviewer #4: This study retrospectively analyzed the clinical data of 555 patients with metatarsal fractures from 2018 to 2022. A model for predicting postoperative functional recovery was constructed and validated for the first time using the LASSO regression method. Ten independent risk factors were identified, including living environment, smoking, obesity, rehabilitation training, education level, age, injury mechanism, infection, and anemia. The performance of the model was evaluated by training cohorts and validation cohorts (C index was 0.832 and 0.838, respectively). The model was eventually developed into a web-based interactive calculator to provide personalized prediction support for patients and doctors. Although the study has limitations such as single-center design and potential selection bias, its large sample size and rigorous statistical methods significantly improve the reliability and clinical application value of the results. The reviewer believes that this study has been modified in accordance with the reviewer's comments.

7. PLOS authors have the option to publish the peer review history of their article (what does this mean? ). If published, this will include your full peer review and any attached files.

**Do you want your identity to be public for this peer review?** For information about this choice, including consent withdrawal, please see our Privacy Policy .

Reviewer #2: No

Reviewer #4: No

---

## [Editor Report · Acceptance letter]

PONE-D-24-51117R3

PLOS ONE

Dear Dr. Lv,

I'm pleased to inform you that your manuscript has been deemed suitable for publication in PLOS ONE. Congratulations! Your manuscript is now being handed over to our production team.

Kind regards,

on behalf of

Professor Yaodong Gu

Academic Editor

PLOS ONE